# A modified *Agrobacterium*-mediated transformation for two oomycete pathogens

**Luyao Wang[1,2], Fei Zhao[2], Haohao Liu[2], Han Chen[2], Fan Zhang[2], Suhua Li[1], Tongjun Sun[1], Vladimir Nekrasov[3], Sanwen Huang[1]\*, Suomeng Dong[2]\***

1 Shenzhen Branch, Guangdong Laboratory of Lingnan Modern Agriculture, Genome Analysis Laboratory of the Ministry of Agriculture and Rural Affairs, Agricultural Genomics Institute at Shenzhen, Chinese Academy of Agricultural Sciences, Shenzhen, China, 2 Department of Plant Pathology and Key Laboratory of Integrated Management of Crop Disease and Pests (Ministry of Education), Nanjing Agricultural University, Nanjing, China, 3 Plant Sciences and the Bioeconomy, Rothamsted Research, Harpenden, United Kingdom

\* huangsanwen@caas.cn (SH); smdong@njau.edu.cn (SD)

## Abstract

Oomycetes are a group of filamentous microorganisms that include some of the biggest threats to food security and natural ecosystems. However, much of the molecular basis of the pathogenesis and the development in these organisms remains to be learned, largely due to shortage of efficient genetic manipulation methods. In this study, we developed modified transformation methods for two important oomycete species, *Phytophthora infestans* and *Plasmopara viticola*, that bring destructive damage in agricultural production. As part of the study, we established an improved *Agrobacterium*-mediated transformation (AMT) method by prokaryotic expression in *Agrobacterium tumefaciens* of *AtVIP1* (VirE2-interacting protein 1), an Arabidopsis bZIP gene required for AMT but absent in oomycetes genomes. Using the new method, we achieved an increment in transformation efficiency in two *P. infestans* strains. We further obtained a positive GFP transformant of *P. viticola* using the modified AMT method. By combining this method with the CRISPR/Cas12a genome editing system, we successfully performed targeted mutagenesis and generated loss-of-function mutations in two *P. infestans* genes. We edited a MADS-box transcription factor-encoding gene and found that a homozygous mutation in *MADS-box* results in poor sporulation and significantly reduced virulence. Meanwhile, a single-copy avirulence effector-encoding gene *Avr8* in *P. infestans* was targeted and the edited transformants were virulent on potato carrying the cognate resistance gene *R8*, suggesting that loss of *Avr8* led to successful evasion of the host immune response by the pathogen. In summary, this study reports on a modified genetic transformation and genome editing system, providing a potential tool for accelerating molecular genetic studies not only in oomycetes, but also other microorganisms.

## Author summary

Oomycetes are a unique group of animal and plant pathogens that include some of the biggest threats to food security and natural ecosystems. However, much of the

**Data Availability Statement:** All relevant data are within the manuscript and its Supporting Information files.

**Funding:** In this research, S.W. and S.D. received funding support from Guangdong Major Project of

Basic and Applied Basic Research (2021B0301030004); S.D. received funding support from National Natural Science Foundation of China (NSFC, 31721004) and China Agriculture Research System (CARS-09-P20); L.W was supported by NSFC (31900303) and the Agricultural Science and Technology Innovation Program (CAASZDRW202101); V.N. was supported by the Biotechnology and Biological Sciences Research Council (BBSRC) through the Designing Future Wheat (DFW) Institute Strategic Programme (grant number BB/P016855/1). The funders had no role in study design, data collection and analysis, decision to publish, or preparation of the manuscript.

**Competing interests:** The authors have declared that no competing interests exist.

pathogenesis and development in these organisms remains to be learned, largely due to shortage of efficient genetic manipulation methods for a long time. In this manuscript, we developed modified *Agrobacterium*-mediated genetic transformation strategies for *Phytophthora infestans*, a notorious oomycete species that caused Irish Famine in the 19th century, and *Plasmopara viticola*, the causal agent of grapevine downy mildew that is known as a highly destructive disease of grapevines in all grape-growing areas of the world. Using prokaryotic expression of a plant protein in *A. tumefaciens*, we achieved considerable increase in transformation rate in different *P. infestans* strains, and also acquired one positive transformant of *P. viticola*, a oomycete species that is extremely hard-to-transform and cannot be grown as an axenic culture. Using our improved transformation protocol, combined with the CRISPR/Cas12a system, we performed genome editing and created loss-of-function alleles in *P. infestans*. In summary, our study reports on modified genetic transformation methods, for two important oomycete species, that have the potential to accelerate the molecular genetic study of many other microorganisms.

## Introduction

Oomycetes are regarded as an important lineage of eukaryotic organisms that includes a considerable number of plant pathogens, which pose a threat to natural and arable ecosystems. Grapevine downy mildew, caused by *Plasmopara viticola*, is considered to be the most devastating disease of grapevines in climates with relatively warm and humid summers. *P. viticola* is an obligate biotrophic oomycete species that cannot be grown as an axenic culture and is very recalcitrant to genetic transformation as demonstrated in previous studies [1]. The genetic transformation obstacle in the case of *P. viticola* has severely hampered functional genomics research and studies on screening molecular drug targets. Thus, it is important to set up a workable transformation method for *P. viticola* and other hard-to-transform biotrophic microorganisms.

Potato late blight is another one of the most well-known but not well-understood plant diseases in terms of molecular genetics of *Phytophthora infestans*, its causal pathogenic microorganism. *P. infestans*, the oomycete pathogen responsible for the devastating potato late blight, poses a worldwide threat [2], and plays an essential role in studying plant–microbe interactions, epidemiology, and field disease control [3]. Currently, reverse genetics studies in *P. infestans* are also hampered by inefficient genetic transformation methods. Up until now, functional genomic research in *P. infestans* has relied mostly on transient/stable gene overexpression or target-gene silencing by RNAi [4, 5]. Although polyethylene glycol (PEG)-mediated transformation, *Agrobacterium*-mediated transformation (AMT), microprojectile bombardment and zoospore electroporation have been already set up in *P. infestans* for years, we need to be open to new methods as other transformation options are always encouraged [6–9]. Moreover, phenotypes of *Phytophthora* transformants often vary significantly between different pathogen lines, experiment operators, and individual experiments, possibly because of random gene integration sites and different silencing or overexpression qualities [10]. Despite of multiple attempts to apply CRISPR/Cas9 for the purpose of targeted mutagenesis in *P. infestans*, no genome editing events, generated using this system, have been reported so far [11]. In a recent study, the CRISPR/Cas12a (Cpf1) system was utilized to produce genome editing events in a *P. infestans* strain using the PEG-mediated protoplast transformation method [12]. To propel future research, it is of great interest to expand the range of available genetic transformation methods for genome editing in *P. infestans*.

*Agrobacterium tumefaciens*, as a typical plant pathogenic bacterium, causes crown gall disease in a wide range of plants [13], and was developed into an efficient genetic transformation tool since Binns and Thomashaw first demonstrated that *A. tumefaciens* can integrate exogenous gene segments into the plant nucleus [14]. Known as the first example of prokaryote to eukaryote horizontal gene transfer, the capacity of *A. tumefaciens* to transfer alien genes to host cells has become the basis of one of the most essential technologies in manifold research fields, and plant science in particular [15]. By now, transgenesis, achieved through *A. tumefaciens*-mediated transformation (AMT), has been established in many model plant species, including Arabidopsis, tobacco and rice [16–18]. In addition to plants, *Agrobacterium* is also able to infect a broad range of other non-plant hosts during the co-cultivation process, including microorganisms identified as plant pathogens, and AMT protocols have been developed as a transformation system for many fungi, including *Aspergilus* species, *Beauveria* species, *Botrytis cinerea*, *Candida* species, *Coccidiodes* species, *Colletotrichum* species, *Fusarium* species, *Trichoderma* species and *Verticillium* species [19–30]. Attempts to improve the efficiency of AMT have been undertaken since the method was first reported. However, in most cases, the efforts were focused on optimization of conditions, such as explant treatment, tissue culture medium composition, temperature or selectable marker, but not the core tool—*A. tumefaciens* [31]. One successful attempt to improve the plant transformation ability of *A. tumefaciens* involved introducing a compatible plasmid carrying a *virG* mutant (*virG*n45D), which constitutively induced *vir* gene expression during AMT [32].

The first AMT protocol for *P. infestans* was established in 2003 by Vijin and Govers; in this study, the *A. tumefaciens* LBA1100 strain was selected, and as many as 30 transformants per $10^7$ zoospores could be obtained [7]. A recent study has also described an efficient AMT protocol for another oomycete species, *Phytophthora palmivora*, which produces large amount of zoospores ($2–5 \times 10^6$/mL) during in vitro preparation [33]. However, the amount of zoospores produced may vary dramatically in different *Phytophthora* strains, and some oomycete species produce hardly any zoospores under artificial conditions. Thus, the ability to produce sufficient amounts of zoospores under in vitro conditions is likely a serious limiting factor for applying zoospore-dependent transformation methods in oomycetes [34–36].

Previous studies illustrated that ectopic expression of several plant proteins significantly improved AMT rate in a range of plant species [37]. A set of proteins from the host are involved with *Agrobacterium*-mediated transformation of plant [37]. Overexpressing several of these host proteins, mostly from *Arabidopsis*, such as AtVIP1, AtRTNLB1, Ku80, histone H2A and SGA1, significantly increases AMT efficiency [37–42]. *Arabidopsis* VirE2-interacting-protein1 (AtVIP1) is known as an important plant protein that contributes to the AMT process [38, 43]. AtVIP1 has been proven to interact with *Agrobacterium* effector VirE2, and its ability to form homomultimeric protein complexes and interact with histone H2A in host cells is required for *Agrobacterium*-mediated pathogenicity or stable genetic transformation [44]. *AtVIP1*-overexpressing plant lines have shown significantly increased susceptibility to *A. tumefaciens* infection, and AtVIP1 seems to facilitate nuclear transport of VirE2 and the T-DNA complex [42]. It would be more practical if we could directly utilize AtVIP1 protein during the AMT process instead of constructing *AtVIP1* transgenic material prior to transformation of a gene of interest.

Here we report on a modified AMT method utilizing AtVIP1 as an enhancer in the transformation of two important oomycete pathogens. In this study, an increment in the transformation efficiency of *Phytophthora infestans* was achieved by prokaryotic expression of AtVIP1 in the *A. tumefaciens* EHA105 strain as compared to AMT without AtVIP1. We then extended application of this modified AMT procedure to *Plasmopara viticola*. By combining our modified AMT method and the recently established Cas12a-based genome editing system, we

successfully generated genome-edited *P. infestans* transformants and observed the expected virulence phenotypes after inoculating potato leaves with these mutants. Our work is providing a new direction for AMT improvement and a way to potentially accelerate molecular genetic studies of the two devastating plant pathogens and other microorganisms.

## Results

### Oomycetes lack homologues of specific plant proteins important for *Agrobacterium*-mediated transformation

To better understand the low AMT efficiency in *P. infestans* and test whether introducing plant proteins into transformation system would significantly improve *Phytophthora* transformation efficiency, we first performed a bioinformatic study of proteins with similarities to previously identified plant proteins, required for AMT, in different oomycete species. We used target protein sequences from *Arabidopsis* to perform local BLAST using publicly available genomic data from each selected plant or oomycete species. Sequence similarity shown in Fig 1A is based on the ratio between the highest alignment score (bit-score) of local BLAST results from a target species and the one from Arabidopsis. In the selected monocotyledonous and dicotyledonous plants, the sequence similarities of the obtained proteins were all above 0.25, except that the sequence similarity of Rad50 in *Solanum tuberosum* and *Vitis vinifera* were merely 0.04 and 0.17, respectively (Fig 1A and S3 Table). In oomycete species, similar proteins with sequence similarity greater than 0.45 were detected for IMPA-4, Rab8, SGA1 and histone H2A. For AGP17, PP2C, Ku70, Rad50 and Ku80, sequence similarities of similar proteins were mostly above 0.15 (average values were 0.19, 0.19, 0.20, 0.18 and 0.15, respectively) in selected oomycete species. Interestingly, sequence similarity of AtVIP1 or its related *Arabidopsis* paralogues, including bZIP29, bZIP30, bZIP69, posF21, bZIP18 and bZIP52, were less than 0.08 in all selected oomycete species, indicating that absence of an AtVIP1 homologue might have a negative impact on the AMT rate in oomycetes.

### AtVIP1 enhances *Agrobacterium*-mediated genetic transformation

To further test our hypothesis that introduction of plant proteins, required for AMT, into a transformation system increases AMT efficiency in oomycetes, we constructed a vector to deliver AtVIP1 from *Agrobacterium* to host cells to potentiate the transformation process. To make AtVIP1 translocatable by *Agrobacterium* T4SS, we fused the coding sequence of AtVIP1 to *virFΔ42N* (truncated *virF*, lacking the sequence encoding the 42 N-terminal amino acids, that functions as a C-terminal transport signal for VirB/D4-translocated proteins in *Agrobacterium* [45]), and added green fluorescent protein-encoding sequence (*GFP*) to the N-terminus (S1A Fig). As shown in S1B Fig, GFP signal was observed in both cytoplasm and nucleus of infected *Nicotiana benthaminana* cells 3 days post leaf infiltration with *A. tumefaciens* EHA105 strain carrying p302b-*gfp-AtVIP1-virFΔN42*. These data suggested that the GFP-AtVIP1-virFΔ42N fusion protein was translocated into infected cells through T4SS during *Agrobacterium* infection.

To test whether our modified protocol enhances AMT efficiency, we first tested it in wheat, a cereal crop with low AMT efficiency [46]. We introduced the prokaryotic expression plasmid p302b-*AtVIP1-virFΔN42* (pV1F) and the binary plasmid pWMB110-*GUS* carrying the β-glucuronidase coding gene *gus* (contains maize *adh1* intron) into *Agrobacterium* strain EHA105 (S2A Fig). By inoculating leaf segments of wheat cv. 'Fielder' with *A. tumefaciens* EHA105 strain containing pV1F and pWMB110-*GUS*, a significantly higher transient transformation efficiency in leaf tissue was observed as compared to the control treatment inoculated with

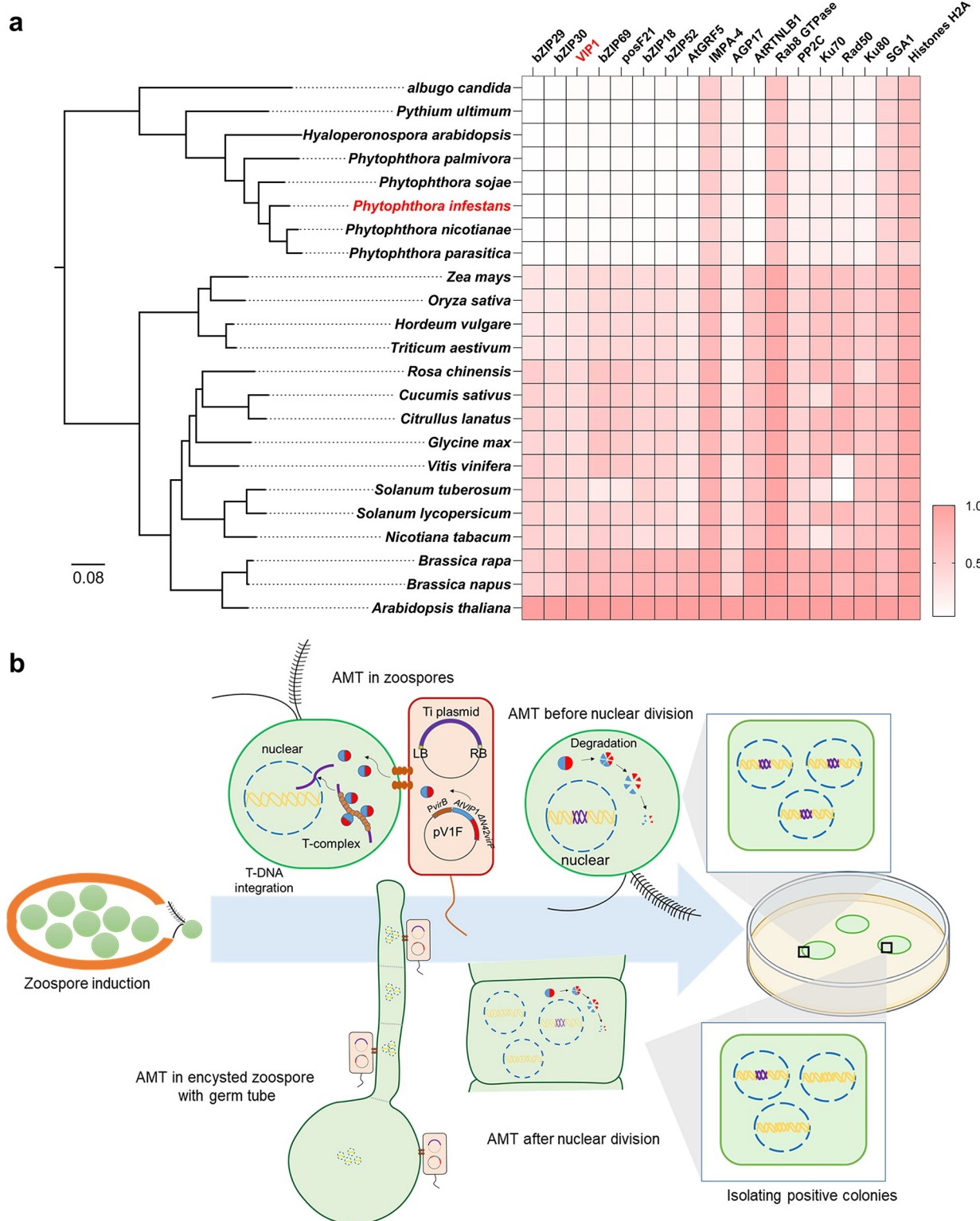

**Fig 1.** (a) The protein similarity analysis of plant proteins, important for AMT of *Arabidopsis*, and their homologues in other plant and oomycete species. Representative plants with an established *Agrobacterium*-mediated transformation protocol and eight oomycete species were selected for phylogenetic analysis. The left phylogenetic tree was generated by the OrthoFinder software based on variations of single-copy orthologous genes in released genomic data of different species using IQ-tree with JTT + G4 model of evolution. The left bar indicates amino acid substitutions per site. The right heatmap indicates sequence similarity of known plant proteins envolved in *Arabidopsis* AMT process and their

homologues in selected plants and oomycete species. The right bar indicates sequence similarity. (b) Schematic representation of the main idea of our improved transformation strategy. The use of the helper plasmid pV1F results in expression of virFΔN42 tagged AtVIP1. The recombinant AtVIP1 is delivered into oomycete zoospores via VirB/VirD4 T4SS, and thereby facilitates the process of T-DNA complex (T-complex) targeting the nucleus of an infected zoospore or an encysted zoospore with a germ tube.

EHA105 strain containing the vector lacking the *AtVIP1* gene (pEV) (S2B Fig). In addition, *gus* expression in root segments increased from 1.8% to 11.6% when transformed with the vector containing the *AtVIP1* gene (S2C and S2D Fig). These results indicate that *AtVIP1* is an effective enhancer of *Agrobacterium*-mediated genetic transformation in plant tissues.

## Expressing *AtVIP1* in *A. tumefaciens* induces T-DNA integration in *P. infestans*

To extend our AMT system to a wider range of eukaryotic species, characterized by low transformation efficiency, we tested it in one of the most important phytopathogenic oomycetes, *P. infestans*, in which AMT protocols have already been set up since years ago [7, 47]. A schematic illustration of our modified AMT method for *P. infestans* is presented in Fig 1B. To test if *AtVIP1* also facilitates AMT in *P. infestans*, we first constructed the binary vector pLY40-*gfp* (Fig 2A), which carries a T-DNA region containing the *gfp* gene driven by *Bremia lactucae Ham34* promoter, and the geneticin (G418) resistance gene *nptII* driven by *Phytophthora sojae Rpl41* promoter. The pLY40-*gfp* was introduced into *A. tumefaciens* strain EHA105 together with either pV1F or pEV.

*A. tumefacien*s strains carrying above-mentioned DNA construct combinations (Fig 2A) were co-cultivated with zoospores of *P. infestans* strains JH19 and HB1501, and subsequently geneticin-resistant transformants were obtained according to the methods presented in the flow diagram in S3 Fig. While G418 resistant colonies were obtained with both tested construct combinations, co-cultivation with *A. tumefaciens* containing the pV1F plasmid resulted in significantly more G418 resistant colonies (or colonies with observed GFP-signal) in both HB1501 and JH19 (Fig 2B–2C and Table 1). One of the representative transformants (JH19 background), named as T1, which was obtained with this modified AMT method, expressed a GFP-size protein and showed a strong GFP signal compared to the transformant T3 (JH19 background) that was obtained using the empty pLY40 as negative control (Fig 2D and 2E). The Southern blot analysis of representative transformants of the HB1501 strain background suggested that T-DNA fragments with the *nptII* gene were integrated into the genomic DNA of all seven of them (Fig 2F). These results indicate that prokaryotic-expressed *AtVIP1* in *A. tumefaciens* considerably promotes T-DNA integration during AMT in *P. infestans*.

## Transformation of *Plasmopara viticola* using the modified AMT procedure

In this study, we also extended application of our modified AMT protocol to *P. viticola*, a difficult-to-transform oomycete that causes grapevine downy mildew. Based on the previously established AMT protocol for the biotrophic fungus *Podosphaera xanthii* [48], *A. tumefaciens* EHA105, carrying LY40-*gfp* and pV1F, was used for co-cultivation with released zoospores of *P. viticola* isolate BS5. After resistance selection, applied by rinsing *P. viticola* inoculated grape leaves with G418, we isolated one transformant called T1 (Figs 3 and S5B). We observed a strong GFP signal in both sporangia and mycelia of T1 during infection using confocal microscopy, and the GFP signal was stable on grape leaves in three sub-inoculation generations (Fig 3A and 3C). Western blot analysis indicated that a GFP-size band could also be detected in protein extracts from the transformant T1 (Fig 3B), and Southern blot analysis, carried out using PCR-amplified *nptII* gene as a probe, showed that one T-DNA segment was successfully

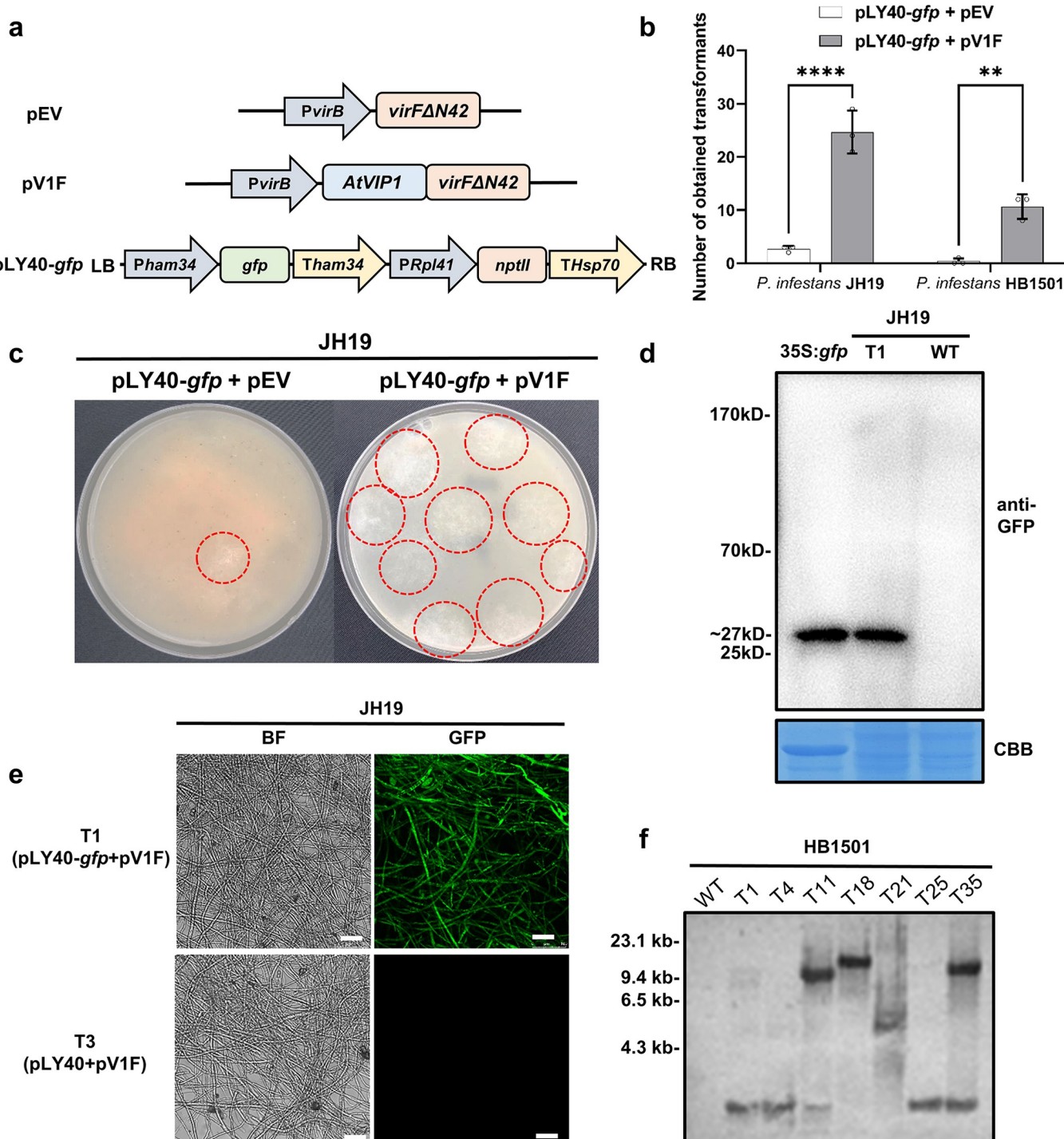

**Fig 2. Generating *P. infestans* transformants expressing *gfp* using the modified AMT method.** (a) Schematic representation of the constructs used in this experiment. *A. tumefaciens* strain EHA105 with pV1F and pLY40-*gfp* was used for *P. infestans* transformation. (b) Quantification of colonies that acquired G418 resistance as a result of transforming *gfp* into *P. infestans* strains JH19 and HB1501 using our modified AMT procedure. EHA105 with the helper plasmid pEV was used as control in this experiment. All data represent average values from three independent experiments with the indicated standard deviations. Statistical differences among the samples were analyzed with Šídák's multiple comparisons test (P< 0.0021: **, P< 0.0001: ****). (c) Two representative plates from the transformation experiment shown in (b). (d) Immunoblot of *P. infestans* JH19 transformant T1, expressing free *gfp*, probed with an anti-GFP antibody. Protein extracted from *N. benthamiana* leaves transiently expressing *gfp* driven by the CaMV35s promoter was used as positive control in lane 1. (e) The *P. infestans* JH19 transformant T1 expressing a detectable GFP signal was obtained by AMT using *A. tumefaciens* carrying constructs described in (a). Scale bars = 40 μm. T3, a randomly selected empty vector transformant was used as negative control. GFP expression in the transformant was analysed by

confocal microscopy. The protein blot was stained with Coomassie Blue to confirm equal loading. (f) Southern blot analysis of representative *gfp* transformants of *P. infestans* strain HB1501. Genomic DNA (4 μg) was digested with *HindIII* and all blots were probed with a fragment containing the *nptII* gene to detect the presence of T-DNA. Numbers on the left indicate the positions of molecular weight markers (kb).

integrated into the genome of T1 (Fig 3D). Consequently, the extensibility of our strategy to *P. viticola* suggests that AtVIP1 could also be considered for optimizing AMT methods in the case of other oomycete species with poor efficiency of transformation.

## AtVIP1 facilitates *Agrobacterium*-mediated delivery of *LbCas12a* into *P. infestans*

To test if our modified AMT method could also facilitate genome editing in oomycetes, we selected *P. infestans* for the first attempt as the Cas12a-based method had recently been set up in this species [12]. We first fused the coding sequences of the *Phytophthora sojae* NLS, human codon-optimized *Lachnospiraceae bacterium* Cas12a (*LbCas12a*) and GFP, and inserted the recombinant sequence into the T-DNA region of pLY40 to obtain pLY40-*Cas12a-gfp* (Fig 4A). *A. tumefaciens* strain EHA105 carrying pV1F and pLY40-*Cas12a-gfp* was used to transform zoospores of *P. infestans* strains JH19 and HB1501. As a result, up to 61 and 22 G418 resistant transformants were obtained for JH19 and HB1501, respectively, in three independent transformation experiments (Table 1). The localization of the GFP signal in the transformants T5 (JH19) and T6 (HB1501) was distinctly nuclear as visualized by confocal microscopy (Fig 4B).

**Table 1. Genetic transformations of *P. infestans* strains JH19 and HB1501 by *A. tumefaciens* EHA105 with or without the helper plasmid pV1F.**

| Binary plasmid | Helper plasmid[a] | *P. infestans* strain | GFP+/G418[R] colonies in attempt 1[b] | GFP+/G418[R] colonies in attempt 2 | GFP+/G418[R] colonies in attempt 3 | Average GFP/G418[R] colonies |
|---|---|---|---|---|---|---|
| pLY40-*gfp* | pEV | JH19 | 0/3 | 1/2 | 1/3 | 0.67/2.67 |
| pLY40-*gfp* | pV1F | JH19 | 11/21 | 10/29 | 13/24 | 11.33*/24.67* |
| pLY40-*gfp* | pEV | HB1501 | 0/0 | 0/1 | 0/0 | 0/0.33 |
| pLY40-*gfp* | pV1F | HB1501 | 2/8 | 7/12 | 5/12 | 4.67*/10.67* |
| pLY40-*Cas12a-gfp* | pEV | JH19 | 1/1 | 0/7 | 0/2 | 0.33/3.33 |
| pLY40-*Cas12a-gfp* | pV1F | JH19 | 13/22 | 3/19 | 6/20 | 7.33*/20.33* |
| pLY40-*Cas12a-gfp* | pEV | HB1501 | 0/0 | 0/0 | 0/0 | 0/0 |
| pLY40-*Cas12a-gfp* | pV1F | HB1501 | 3/6 | 3/7 | 7/9 | 4.33*/7.33* |
| Binary plasmid | Helper plasmid | *P. infestans* strain | G418[R] colonies in attempt 1 | G418[R] colonies in attempt 2 | G418[R] colonies in attempt 3 | Average G418[R] colonies |
| pLY40-*PiMADS*-KO | pEV | JH19 | 0 | 0 | 1 | 0.33 |
| pLY40-*PiMADS*-KO | pV1F | JH19 | 7 | 4 | 5 | 5.33* |
| pLY40-*Avr8*-KO | pEV | HB1501 | 2 | 2 | 1 | 1.67 |
| pLY40-*Avr8*-KO | pV1F | HB1501 | 11 | 9 | 8 | 9.33* |

[a] Helper plasmids used in this experiment are described in detail in S2 Table.

[b] As determined with GFP-observation by confocal microscopy (GFP).

As determined 14 days after acquired transformants transferred to another rye-sucrose medium with 5 or 10 μg/L geneticin (G418).

*The marked values indicate significantly different with the AMT process used same binary plasmid but pEV as the helper plasmid. Student's t-test was used to determine the differences.

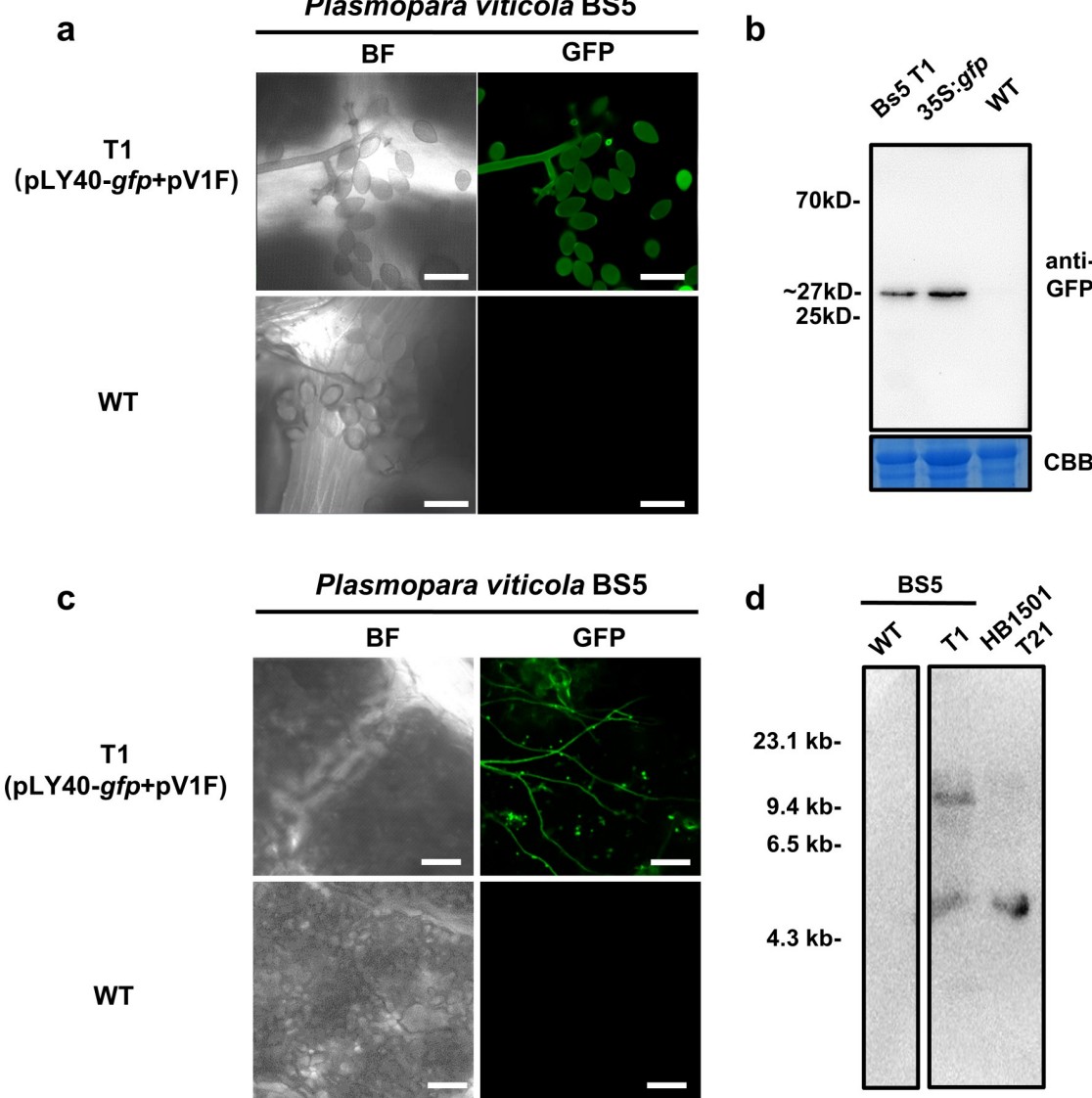

**Fig 3. Obtaining *Plasmopara viticola* BS5 stable transformant T1 using the modified AMT protocol.** (a) *P. viticola* BS5 strain was used for transformation assay and was obtained by AMT using *A. tumefaciens* carrying constructs pLY40-gfp and pV1F. Scale bars = 40 μm. The confocal microscopy images were taken 7 days post inoculation with the zoospores from the third sub-generation of T1; wild type BS5 was used as negative control. (b) Immunoblot of *P. viticola* BS5 transformant T1 expressing free GFP, probed with an anti-GFP antibody. Protein extracted from *N. benthamiana* leaves transiently expressing *gfp* driven by the CaMV35s promoter was used as positive control in lane 2. (c) Transformant T1 of *P. viticola* BS5 expresses detectable GFP signal when infecting grapevine leaves (Zitian seedless, A17). Scale bars = 100 μm. Images were taken 7 days post inoculation and wild type BS5 was used as negative control. (d) Southern blot analysis of transformant T1 of *P. viticola* BS5. Genomic DNA (4 μg) was digested with *HindIII* and all blots were probed with a fragment containing the *nptII* gene to detect the presence of T-DNA. Numbers on the left indicate the positions of molecular weight markers (kb).

As shown in Fig 4C, the hlbCas12a-GFP fusion protein was successfully detected by western blotting in both transformants T5 and T6 with *hlbCas12a-gfp* expressed in *N. benthamiana* used as a positive control. These data suggest successful delivery of the *Cas12a* gene into *P. infestans* using our modified AMT method, paving a way for an efficient genome editing method for this species.

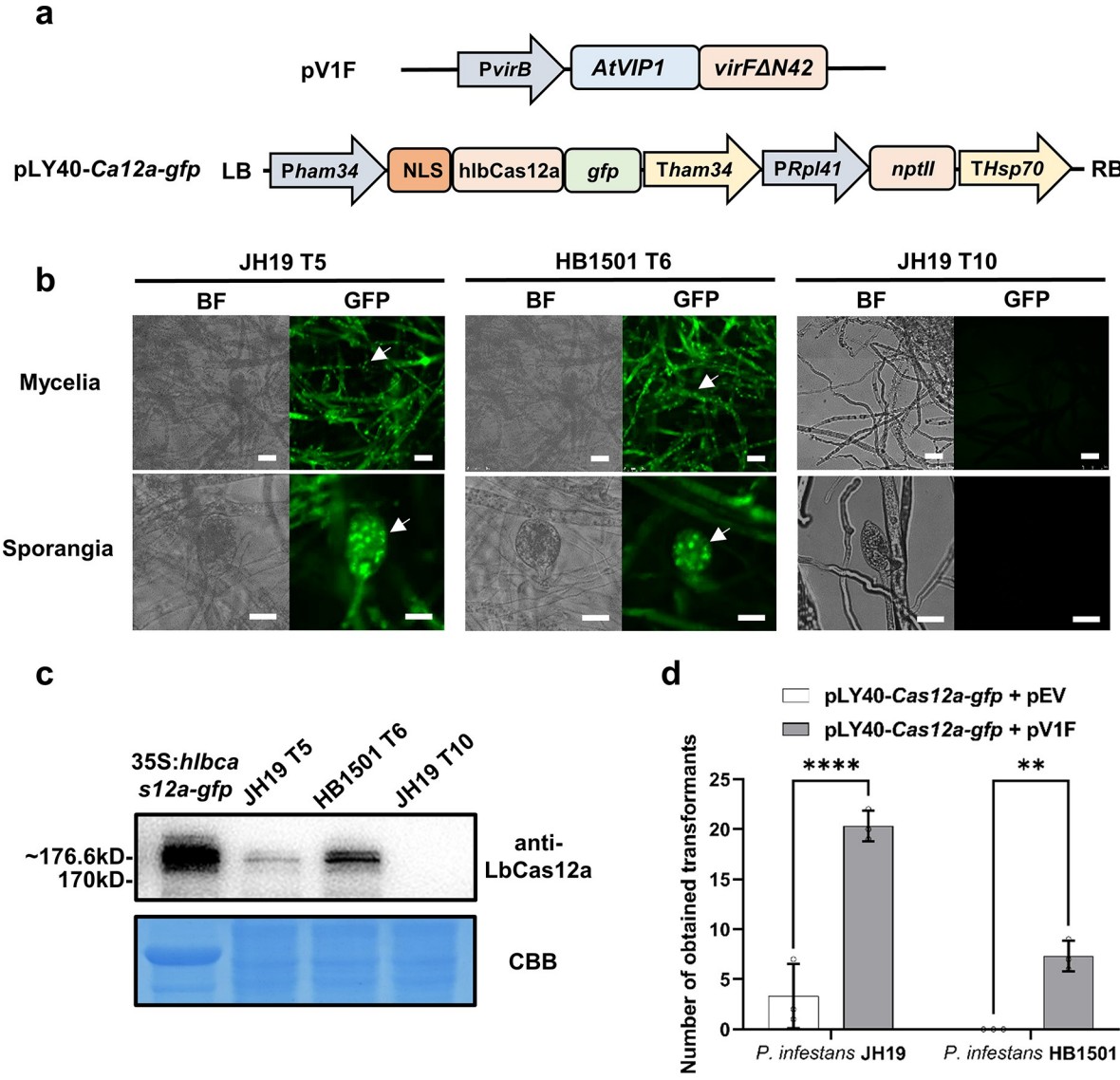

**Fig 4. Integration of *LbCas12a* in *P. infestans*.** (a) Schematic representation of the constructs used in the experiment. *A. tumefaciens* EHA105 carrying pLY40-*Cas12a-gfp* and either pV1F or pEV was used for *P. infestans* transformation. (b) Confocal micrograph of T5 and T6 transformants expressing PsNLS-lbCas12-GFP. The nuclear localization pattern of the fusion protein, indicated by white arrows, revealed mycelia and sporangia. T3, an empty vector pLY40 transformed line, is used as negative control. Bright field and GFP channels are presented. Scale bars = 40 μm. (c) Immunoblot of two representative *P. infestans* transformants expressing PsNLS-lbCas12-GFP. The *gfp* tagged *hlbCas12a* driven by CaMV35S promoter was expressed in *N. benthamiana* as positive control. The expected size of the protein is 176.6 kDa. The protein blot was stained with Coomassie Blue to confirm equal loading. (d) Quantification of positive *P. infestans* transformants expressing GFP tagged Cas12a (JH19 and HB1501 backgrounds) generated using the modified AMT procedure. *A. tumefaciens* EHA105 carrying the helper plasmid pEV was used as control in this experiment. Statistical differences among the samples were analyzed with Šídák's multiple comparisons test (P< 0.0021: **, P< 0.0001: ****).

### Editing a MADS-box-encoding gene in *P. infestans*

To investigate the prospect of using the modified AMT method for CRISPR/Cas12a-mediated *P. infestans* genome editing, a single-copy gene encoding a *MADS-box* transcription factor (PITG_07059) was chosen as the first editing target in this study. The homologous *MADS-box* genes play an essential role in asexual reproduction and zoosporogenesis in both *P. infestans* and *P. sojae* [49, 50]. Two gRNAs, MADS-g1 and MADS-g2, targeting the 1332 nt *MADS-box-*

encoding sequence were designed using the procedure described by Ah-Fong et al. (Figs 5A and S6) [12]. In the binary construct pLY40-*MADS-box*-KO, MADS-g1 and MADS-g2 were flanked by 21-bp short direct repeats (DRs) and inserted in a tandem downstream of *LbCas12a* and the 73-nt poly-adenine sequence (pA), which was added to promote translation of Cas12a mRNA (S6C Fig). As a 3' uridine-rich tail positively regulates CRISPR/Cas12a crRNA formation, four thymines were added at the 3'ends of sequences encoding the gRNAs (S6C Fig) [12, 51]. The editing events within the *MADS-box* gene in the transformants were detected by performing PCR, with primers amplifying the full length of the gene, and sequencing the PCR amplicons. In 2 out of 16 G418 resistant transformants (T8 and T16), shorter PCR amplicons were observed suggesting presence of CRISPR/Cas12a-induced deletions (Fig 5B). Sequencing analysis confirmed that T8 and T16 carried identical 993 bp deletions, between MADS-g1 and MADS-g2 loci, resulting in a truncated *MADS-box* gene coding sequence (Figs 5C and S6E). The MEF2-like domain of the MADS-box protein contributes to the DNA binding activity, which is essential for a transcription factor. The MEF2-like domain was predicted to be located at the 2–72 aa sites by InterproScan (version 5.52–86.0) (Fig 5A). The sequencing data presented in Figs 5C and S6E showed that both mutant *MADS-box* alleles, present in T8 and T16, lack the majority of the sequence encoding the MEF2-like domain, suggesting loss of function of the MADS-box proteins in these transformants.

As shown in Fig 5D and 5F, *MADS-box*-edited JH19 strains T8 and T16 showed merely no sporangia output after 5 days of cultivation in PEA broth compared with untransformed wild type JH19, which yielded 49.22 sporangia in 10 μL PEA broth culture. Another transformant (T5) was selected as a non-edited control, in which no significant reduction of sporangia output was detected (Fig 5D and 5F). To check whether the phenotype observed in mutant T8 and T16 strains was specific to sporangia production, we decided to measure the vegetative mycelia growth rate in them. As a result, both T8 and T16 showed similar vegetative growth rates, comparable to the controls, on the rye-sucrose medium at 18°C (S8A and S8B Figs). In addition, we performed an *in planta* test by inoculating potato leaves (*Solanum tuberosum* cv. Désirée) with T5, T8, T16 and wild type JH19. As *MADS-box*-edited JH19 strains yielded no sporangia, we selected mycelial discs instead of zoospores for inoculation assays in this experiment. T5 and wild type JH19 strains caused complete infectious lesions and developed obvious aerial mycelia. In contrast, T8 and T16 only showed partial watery lesions without formation of aerial mycelia (Figs 5E, 5G and S8C). We further examined the disease area by microscopy: both T5 and wild type JH19 strains produced sporangia in heavily diseased potato leaves, while T8 and T16 only developed sparse mycelia without detectable sporangia (S8C Fig).

## Editing Avr8-encoding gene in *P. infestans* HB1501

Phytophthora avirulence (Avr) genes are key determinant factors for gene-for-gene interaction with host plants such as potato and soybean (Dong et al., 2011). We selected *Avr8* (PITG_07558), known as an avirulence gene recognized by potato late blight resistance gene *R8* [52], as the second editing target. We selected *P. infestans* strain HB1501 for *Avr8* editing since JH19 naturally overcomes the *R8*-mediated resistance, as was previously established using inoculation assays. As shown in S7A Fig, genome-wide single nucleotide polymorphism (SNP) analysis revealed that HB1501 is diploid. *Avr8* encodes a 245 aa RxLR effector and was confirmed as a single-copy gene in strain HB1501 based on the read depth analysis (S7B Fig). The Avr8 protein includes 3 LWY motifs (63–107, 105–162, 165–218 aa), which might contribute to its novel activities as an RxLR effector during infection progress (Fig 6A) [53]. Cas12a guides RNAs Avr8-g1 and Avr8-g2 were designed to target *Avr8* before the first LWY motif to increase chances of introducing a frame-shift mutation before this motif due to

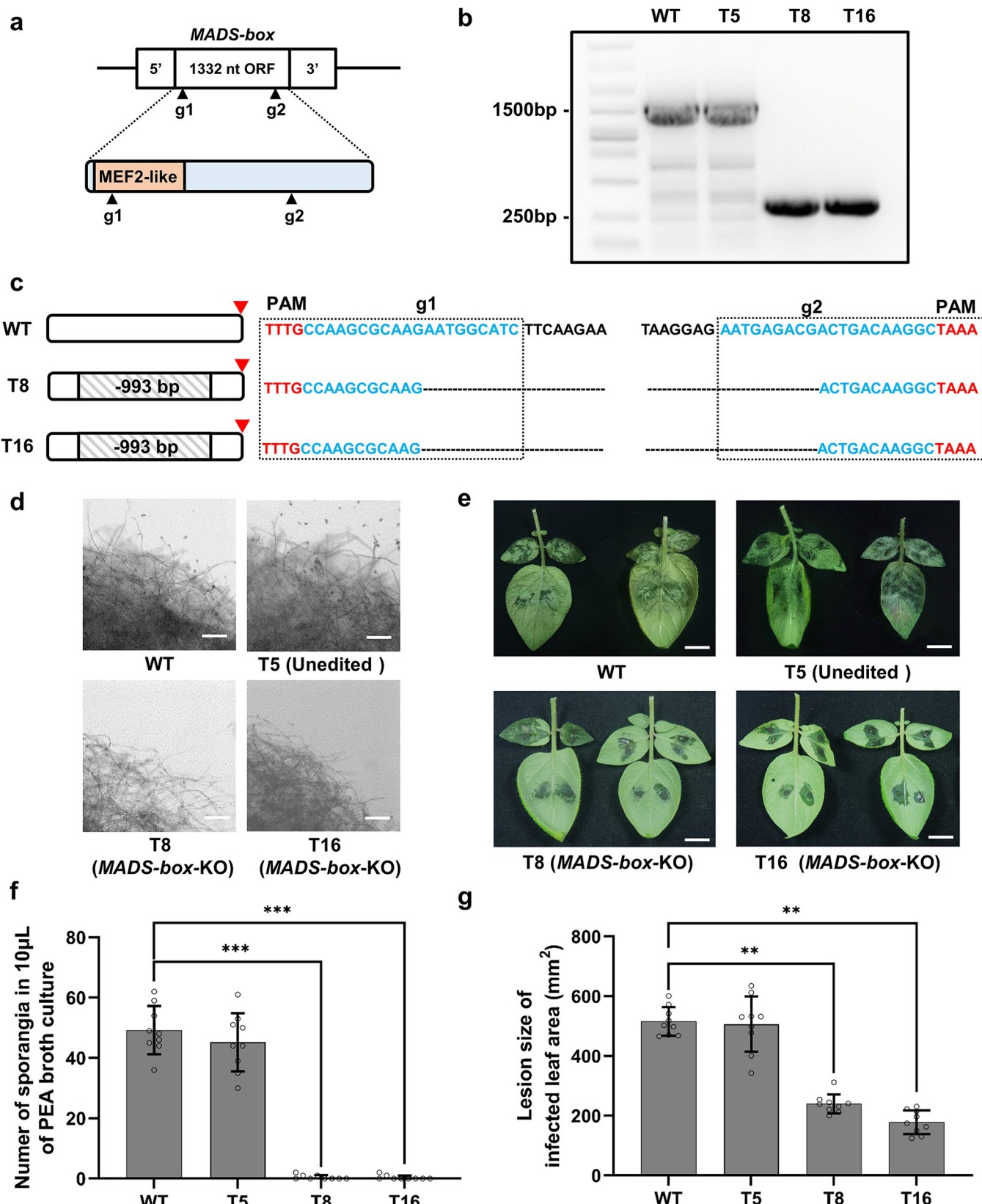

**Fig 5. Editing a MADS-box transcription factor coding gene in *P. infestans* strain JH19.** (a) Target sites of two gRNAs in the *MADS-box* coding sequence. The MEF2-like domain was predicted by InterProScan (version 5.52–86.0). The MEF2-like domain locates at 2–77 aa sites. (b) Detecting of editing events in *MADS-box* in JH19 transformants. The PCR assay on T8 and T16 revealed two homozygous editing events. (c) Analysis of PCR amplicon sequencing results from (b). The PAM sequences are marked in red, the target sequences are marked in blue, and the stop codons are indicated by red arrowheads. Each dash line represents a deleted nucleotide. (d) Culture scrapings from 10 days Pea broth cultures of wild type JH19, T5, T8 and T16. Scale bars = 0.4 mm. (e) Infection phenotypes of wild type JH19 (WT), transformants T5, T8 and T16 on leaves of susceptible cultivar potato cv. Désirée. Detached potato leaves were inoculated with mycelia medium discs, and images were recorded 5 days post inoculation. Scale bars = 2 cm. (f) Quantification of sporangia numbers in 10 μL PEA broth culture of wild type JH19, T5, T8 and T16 in (d). (g) Quantification of lesion size in detached leaves of potato cv. Désirée inoculated with wild type JH19 (WT), T5, T8 and T16 in (e). All data represent average values from three independent experiments with the indicated standard deviations. Statistical differences among the samples were analyzed with Šídák's multiple comparisons test (P< 0.0021: **, P< 0.0002: ***).

editing events (S7D Fig). The cartoon illustrating the design of the *Avr8* CRISPR/Cas12a knockout construct is shown in S7C Fig, with Avr8-g1 and Avr8-g2 targets shown in red in S7D Fig.

The transformants were PCR-genotyped for edits in *Avr8* using primers amplifying the full-length gene, with PCR amplicons being subsequently sequenced. Out of 27 G418 resistant transformants, variant bands were observed in two of them, T3 and T10, suggesting both *Avr8* alleles were altered in them (Fig 6B). Sequencing of the PCR amplicons from the shifted T3 and T10 bands showed deletions spanning both Avr8-g1 and Avr8-g2 target sites within *Avr8*. Unlike the editing events detected in *MADS-box*, edits in the *Avr8* gene in T3 and T10 resulted in frame-shift mutations predicted to cause early termination of protein translation before the first LWY motif (Fig 6C).

As a following step, we performed virulence assays by inoculating detached leaves of *R8* transgenic potato (Désirée *R8*) with transformants T3, T10, T22 (unedited) and wild type HB1501 strain. We recorded the infection phenotypes 5 days post zoospore inoculation. Compared with unedited T22 and wild type HB1501, T3 and T10 caused lesions of significantly larger size upon infection of *R8* transgenic potato leaves. Importantly, both T3 and T10 caused infection symptoms similar to T22 and wild type HB1501 when using detached leaves from susceptible wild type potato (Désirée WT) lacking the *R8* gene (Fig 6D and 6E). We therefore reason that creating a loss-of-function *Avr8* allele enables *P. infestans* to evade *R8*-mediated host resistance.

## Discussion

As a widely used approach for transient and stable expression of exogenous genes, AMT has been set up for a broad spectrum of biological categories, including plants, microorganisms and even human cells [20, 54–56]. The AMT protocols for oomycete species, the majority of which are classified as destructive plant pathogens, have been reported, including *Phytophthora palmivora*, *Phythium ultimum* and *Phytophthora infestans* [57]. Although these AMT methods have been utilized for years, they require optimization due to generally low transformation efficiencies. *Arabidopsis* bZIP family transcription factor AtVIP1, known as a binding partner of *A. tumefaciens* effector VirE2, facilitates VirE2 nuclear import and *A. tumefaciens* infectivity [43, 58]. Moreover, significant improvement of transformation was observed in an *A. tumefaciens* transformation assay using *AtVIP1*-overexpressing tobacco plants [42]. Until now, utilization of *AtVIP1* as a factor boosting the AMT efficiency required two steps: (1) generating a stable *AtVIP1* transgenic line; (2) transforming a gene of interest into the *AtVIP1* transgenic line. Overexpressing *AtVIP1* in a stable transgenic line might cause an unpredictable pleiotropic phenotype as well as limit the range of selectable markers available for transformation of a gene of interest. In addition, when applying *AtVIP1* the way described above, one is presented with a chicken-and-egg problem in the case of plant genotypes, which are not amenable to AMT to start with. A recent study in wheat reported that co-

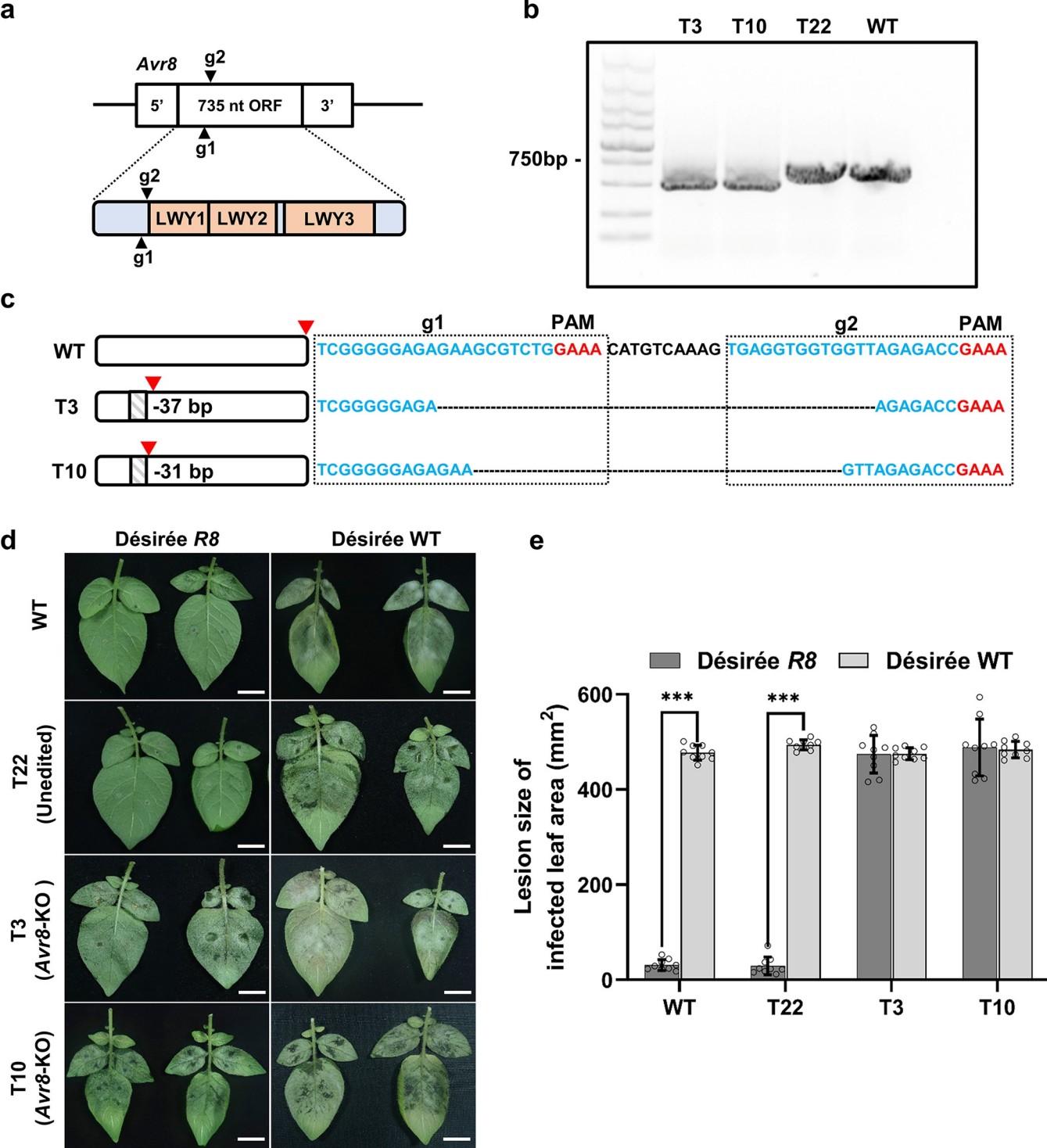

**Fig 6. Editing an avirulence gene *Avr8* in *P. infestans* strain HB1501.** (a) Target sites of two gRNAs in the *Avr8* coding sequence. The LWY motifs were predicted by InterproScan (version 5.52–86.0). The LWY1 domain locates at 63–107 aa sites, the LWY2 domain locates at 105–162 aa sites, and the LWY3 domain locates at 165–218 aa sites. (b) Detecting editing of *Avr8* in HB1501 transformants. The PCR assay in T3 and T10 revealed two homozygous editing events. (c) Analysis of PCR amplicon sequencing results from (b). The PAM sequences are marked in red, the target sequences are marked in blue, and stop codons are indicated by red arrowheads. Each dash line represents a deleted nucleotide. (d) Infection phenotypes of wild type HB1501 (WT), T3, T10 and T22 on detached leaves of *R8* transgenic potato. Detached potato leaves were inoculated with zoospores of selected strains, and images were taken 5 days post inoculation. Scale bars = 2 cm. (e) Quantification of lesion sizes in (d). All data represent average values from three independent experiments with the indicated standard deviations. Statistical differences among the samples were analyzed with Šídák's multiple comparisons test (P< 0.0002: ***).

transformation of the wheat gene *TaWOX5* from the WUSCHEL family with a gene of interest dramatically increases transformation efficiency, resulting in a lower genotype dependency, in 29 wheat varieties. The latter was similar to our original idea of modifying AMT for oomycete species [59]. Here, we present an alternative method utilizing AtVIP1 to optimize AMT for two oomycete species using prokaryotic expression of *AtVIP1* fused with a sequence encoding a T4SS translocation tag in *A. tumefaciens*.

In addition to utilizing AtVIP1, we have done additional modifications to the procedure while developing our modified AMT protocol for *P. infestans* (S3 Fig). Murashige and Skoog (MS) medium is commonly used for co-cultivation of *A. tumefaciens* and the explant material [60, 61]. Considering that *A. tumefaciens* shows a better growth rate in MS medium than in induction medium (IM), we used MS medium instead of IM for co-culturing *A. tumefaciens* and *P. infestans* zoospores. The germination of zoospores and transformation efficiency were not affected by this medium substitution. We used geneticin G418 as a selection agent in this study, while different *P. infestans* wild-type strains showed variation in G418 tolerance. For the JH19 strain, we used 5 μg/mL of G418 for selection of positive transformants, while for the HB1501 strain—10 μg/mL.

In previously reported PEG-mediated *P. infestans* transformation or genome-editing procedures, several different situations might occur upon plasmid introduction: (a) the plasmid might integrate into an unstable site in the transformant's genome; (b) the plasmid DNA might be later degraded by enzymes inside *P. infestans* cells [62]. The modified AMT procedure presented in this study has the advantages of the previously described AMT protocol, including no need to produce protoplasts, no need for a large amount of high-quality plasmid DNA, and, in addition, the integrated T-DNA fragment in the genomic DNA might be more stable than plasmid DNA [63]. The AMT method also comes with its issues e.g. during the T-DNA integration step: (a) some transgene instability might be observed, probably due to rearrangement of the T-DNA region, and/or due to homologous recombination between copies of the transgene inserted into the DNA in the same nucleus; (b) only part of the T-DNA might integrate into genomic DNA [64].

Importantly, during PEG-mediated transformation, the plasmid DNA might not integrate straightaway in protoplasts but at a later stage, during cell differentiation. During the AMT process, T-DNA integration also potentially occurs in encysted zoospores that is followed by emergence of germ tubes. Both of the described above situations would result in chimeric mycelia in resulting transformants, although there are currently no data to indicate that this is the case with AMT [65].

In this study, each AMT experiment started with approximately $8 \times 10^5$ zoospores, and 9 of 18 experiments, conducted using our modified AMT method, produced at least 10 G418-resistant colonies; 5 of 18 attempts produced at least 20 G418-resistant colonies (Table 1). However, further characterization of acquired transformants revealed that part of the isolates, transformed with *gfp*, that showed resistance to G418, did not show a GFP signal during confocal microscopy analysis or expressed a GFP-size protein detectable by western blotting (Table 1 and S4 Fig). The Southern blot analysis of representative transformants of the HB1501 background suggested that the T-DNA segments with the *nptII* gene were integrated into all six selected isolates, which were positive for the GFP signal, and one transformant (T1) with no GFP signal (Fig 2F). The latter could be due to the issue (b) of the AMT protocol that was mentioned above.

During the Southern blot analysis, we observed bands of similar small size (< 4.3 kb) in lanes 2, 3, 4, 7 and 8 (Fig 2F). These results might be caused by non-specific hybridization or, possibly, partial T-DNA integration events in selected HB1501 transformants. The above-mentioned observations are consistent with a similar phenomenon observed during plant

transformation [66]. To characterize better the integration events in transformants of oomycete species, we would like to perform next generation sequencing (NGS), that represents a highly sensitive approach to detect T-DNA insertions in transgenic isolates.

There are few more differences between the PEG-mediated transformation and AMT methods, e.g. the different recipient cell types. Importantly, many protoplast cells used for PEG-mediated transformation are coenocytic, while most zoospores used for AMT are single-nucleated and wall-less [63, 67]. However, we have no sufficient evidence to suggest that AMT of *P. infesteyotic transfor* zoospores could eliminate the issue of potential heterokaryoteyotic transformants, particularly because it was reported that a certain fraction of *P. infestans* zoospores had been observed to be multinucleate [68]. Additionally, the conditions used in our modified AMT method could not eliminate the possibility of encystment of the zoospores followed by emergence of a germ tube during co-cultivation of zoospores and *Agrobacterium* cells. To illustrate such scenario, we included the potential situation of T-DNA integration in multinucleate cells in Fig 1B. We also performed a comparison between the published PEG-mediated transformation and AMT protocol, and our modified AMT protocol for *P. infestans* (S4 Table). As a result, we would define our modified AMT protocol as an option to be considered, rather than top choice, for *P. infestans* transformation, as transformation efficiencies associated with different protocols are difficult to compare due to different standards used.

In addition to *P. infestans*, genetic transformation methods have not been set up for many biotrophic oomycetes. Martínez-Cruz et al. has reported an AMT method for *Podosphaera xanthii*, a biotrophic fungus that causes cucurbit powdery mildew [48]. Based on the AMT method for *P. xanthii*, we extended our modified AMT protocol to *Plasmopara viticola* and successfully obtained a positive transformant (Figs 3 and S5). There are a few more issues that need to be addressed when it comes to AMT of *P. viticola*, including (a) transformation efficiency might vary in different *P. viticola* isolates; (b) leaves from different grapevine varieties may contribute differently to screening of resistant transformants; (c) it would be difficult to recover *P. vitcola* transformants from infected grapevine leaves without aseptic conditions; (d) stable expression of an exogenous gene needs further validation in subsequent generations of obtained transformants [1]. Although we have acquired one positive stable transformant of *P. viticola* based on our modified AMT method with AtVIP1, we still need to do more transformation attempts and analysis to draw a solid conclusion, in respect to the transformation efficiencies, in the future. Surely, we know the importance of acquiring a stable transformant of *P. viticola*, while our experimental procedure comes with a few shortcomings: (a) the transformation efficiency is not high enough as compared to other oomycete species; (b) we have not performed a sufficient number of transformation assays with other *P. viticola* isolates. Consequently, the extensibility of our strategy to *P. viticola* suggests that AtVIP1 could be considered for modifying AMT methods for other oomycete species, especially some hard-to-transform biotrophic oomycetes, such as *Bremia lactucae*.

Unlike gene overexpression, integration of a CRISPR/Cas cassette in transformants does not guarantee successful editing of a gene target. Previous reports have demonstrated that editing efficiency varies greatly, from 1% to 100%, among different eukaryotic pathogens [69]. The editing frequency of CRISPR/Cas12a in *P. infestans* reached 13% by using the PEG transformation method [10, 12]. In this study, editing of the *MADS-box* gene resulted in two homozygous mutants out of 16 transformants, and editing of *Avr8* genes resulted in two homozygous mutants out of 28 transformants (Table 1). Although introducing *AtVIP1* does not seem to increase the frequency of CRISPR/Cas12a-mediated genome editing events in *P. infestans*, our modified AMT method still provides a valuable option for future related studies.

We selected two target genes for CRISPR/Cas12a-mediated genome editing in this study based on two criteria. The first is avoiding multi-copy target genes to reduce the difficulty of

editing and transformant screening, and the second is that transformants with successful editing events should present explicit phenotypes. The cycle of aerial asexual sporangia dispersal plays a crucial role in late blight development [70]. Thus, we selected *MADS-box* as our first target gene that is reported to express only in sporulating mycelia and spores of *P. infestans*. *MADS-box* (PITG_07059) was first identified by Leesutthiphonchai and Judelson in 2018 and MADS-box transcription factors play significant roles in eukaryotes [50]. We utilized modified AMT and CRISPR/Cas12a-mediated genome editing methods and obtained two edited *MADS-box* mutants (Fig 5). Consistently with the results obtained using the RNAi approach, our edited *MADS-box* mutants produced no sporangia and showed reduced ability to infect potato leaves (Fig 5). RNAi is triggered by siRNAs whose silencing efficiency is not guaranteed and varies widely in different transformants. In contrast to RNAi methods, genome editing would provide more stable phenotypic data for further research on *P. infestans*. Since during CRISPR/Cas12a-mediated genome editing, based on the AMT method, the *Cas12a* expression cassette stably integrates into the *P. infestans* genome, an additional self-fertilization step could be used to remove *Cas12a* from the original transformant, similarly to the situation *in planta* [71].

Disrupting the recognition of phytopathogen avirulence (AVR) genes by plant resistance (R) genes normally produces easily observable phenotypes. Thus, we selected *Avr8* (also called *AVRSmira2* and PITG_07558), a single-copy AVR gene recognized by potato late blight resistance gene *R8*, as our second genome editing target. We chose *P. infestans* strain HB1501 for *Avr8* gene editing because seemingly the JH19 strain broke down the *R8*-mediated resistance in potato. *Avr8* was identified by analysis of variance (ANOVA) using the average AUDPC values of both its responses to the *R* gene and field trials [72]. The broad spectrum late blight resistance gene *R8* that recognizes *Avr8* was cloned from *Solanum demissum*, based on a previously published coarse map position on the lower arm of chromosome IX, and the correlation between the expression levels of *Avr8* and *R8*-mediated resistance had been proven in a previous study [73]. However, an inoculation assay of an *Avr8* knockout *P. infestans* mutant on potato carrying the *R8* gene has not been reported yet. In our study, we produced two genome-edited *Avr8* mutants of *P. infestans* (strain HB1501), and both mutants induced infection lesions in leaves of the *R8* transgenic potato line cv. Désirée (Fig 6D and 6E).

Collectively, generating *P. infestans* transformants or genome edited mutants via our modified AMT procedure is less laborious and results in an acceptable transformation rate as compared with the previously reported AMT protocol and PEG-mediated protoplast transformation method. Successfully acquiring a stable transformant of *P. viticola* gives a strong hint of gaining a potential advantage by utilizing proteins, such as AtVIP1, that play an important role in host cells during the AMT process. Although AMT has already been established in plenty of phytopathogen species, it would be particularly interesting to investigate whether our modified AMT protocol would increase the efficiency of transformation or that of CRISPR/Cas-mediated genome editing in them or even extend the effect to other oomycete or plant species.

## Materials and methods

### Growth conditions for *P. infestans*, *P. viticola*, bacteria and plants

*P. infestans* strains were cultured on rye-sucrose medium (agar 15g/L) at 18˚C. *P. infestans* strain JH19 was isolated from infected tomato in San Diego Country, California in 1982 and kindly provided by Howard S. Judelson lab [12]. *P. infestans* strain HB1501 was isolated from Hebei Province in 2015 [74]. *Plasmopara viticola* isolate *BS5* was kindly provided by Dr. Linfei Shangguan at Nanjing Agricultural University and was isolated in Nanjing, China in 2017. *A*.

*tumefaciens* strains and *E. coli* strains were grown on Luria–Bertani (LB) agar (NaCl 10g/L, Yeast extract 5g/L, Tryptone 5g/L, agar 15g/L) at 28°C and 37°C, respectively. *Solanum tuberosum* L. cv. Désirée (2n = 4x = 48) and trangenic lines were grown in soil or in MS medium (MES 0.5 g/L, sucrose 20 g/L, agar 8 g/L, pH5.8), after seed surface sterilization, and maintained in vitro. *Vitis vinifera* (Cultivar grape, Zitian Seedless, A17) was used for culturing of *P. viticola* isolate *BS5*. All plants were grown in environment-controlled growth chambers under long-day conditions (16 h light/8 h dark cycle at 140 μE sec$^{-1}$m$^{-2}$ light intensity) at 22°C.

## Construction of plasmids

Primer sequences used in these cloning procedures are described in S1 Table, and plasmids and cloning strategies are summarized in S2 Table. For *AtVIP1* gene expression in *A. tumefaciens*, the coding sequences of *AtVIP1* (AT1G43700) and *virF* (NC_003065.2) without the N terminal 42 amino acids (*ΔN42virF*) were PCR-amplified, using *A. thaliana* Col-0 cDNA library and *A. tumefaciens* C58 gDNA, respectively, as templates and cloned into p533BL. A pCB302B backbone construct with the *virB1* promoter from *A. tumefaciens* C58 was used to drive acetosyringone induced gene expression in *A. tumefaciens*.

To construct binary plasmid for *P. infestans* transformation, *Bremia lactucae Ham34* promoter and terminator were PCR-amplified and cloned into I-CeuI site of pPZP-RCS2, while *nptII* expression cassette driven by *Phytophthora sojae RPL41* promoter and *B. lactucae* Hsp70 terminator was inserted into AscI site of pPZP-RCS2, which resulted in pLY40. The coding sequence of *gfp* was PCR-amplified and cloned into I-CeuI site of pLY40 separately to obtain pLY40-*gfp*. The sequences of the *RPL41* promoter from *P. sojae*, *Ham34* promoter and terminator, *Hsp70* terminator from *B. lactucae* were all PCR-amplified from pYF515, a plasmid construct used for genome editing in *P. sojae* [75].

To construct binary plasmids for CRISPR/Cas12a mediated *P. infestans* genome editing, the coding sequences of NLS derived from a *P. sojae* bZIP transcription factor [10], human codon-optimized *LbCas12a* from p33lb [76] and artificial synthetic polyA-crRNA-Ham34 terminator segment (S6C and S7C Figs) were fused and cloned into I-CeuI/PacI sites in pLY40, to obtain pLY40-*MADS-box*-KO and pLY40-*Avr8*-KO.

The plasmid sequences of pLY40, pLY40-*gfp* and pLY40-*Cas12a-gfp* are presented in Appendix S1.

## Transient transformation assays in wheat tissue

Transient expression assays in wheat tissue were performed as previously described for *N. benthamiana* with some modifications [77]. *A. tumefaciens* overnight culture was diluted in LB liquid medium without antibiotics, grown for 3–4 h and adjusted to OD$_{600}$ = 0.5. Leaves and roots from 4-weeks old in vitro wheat plant were collected and divided into 5 mm segments, then immersed for 10 min in *Agrobacterium* suspension, placed on MS medium at 22°C for 3 days in growth chamber. Leaf and root segments were then rinsed by sterilized water and transferred into GUS staining solution and incubated at 37°C overnight [78].

## *Agrobacterium* meditated transformation of *P. infestans* zoospores

AMT of *P. infestans* zoospores followed previous published method with few modifications and summarized in S3 Fig [7]. Briefly, *P. infestans* strains were ready for zoospore-induction after 14 days of growth on solid rye-sucrose medium (90mm petri dish plates). Approximately 5 mL of ice-cold sterilized water was used to flush and soak *P. infestans* culture and the culture plates were plated at 4°C for 2 hours, and zoospores will be released (100 zoospores counts/μL for isolate JH19, 200 zoospores counts/μL for isolate HB1501). *A. tumefaciens* strains with

constructed plasmids for transformation was growing overnight with proper antibiotics (25 µg/mL rifampicin, 50 µg/mL kanamycin, 100 µg/mL spectinomycin), 1 mL of *Agrobacterium* culture was added into 50 mL LB liquid medium and grown for 3–4 h till $OD_{600}$ reach 1.0. *Agrobacterium* cells were centrifuged at 4000 rpm for 10 min to collect cells and resuspended by MS liquid medium with 200 µM acetosyringone. *Agrobacterium* suspension was cultured at room temperature for 1 h and then mixed with harvested fresh *P. infestans* zoospores (about $8 \times 10^5$ zoospores for both isolate JH19 and HB1501). The mixture was cultured at room temperature for 45 min and zoospores were collected by centrifuge at 265 g for 5 min, each 200 µL of zoospore culture was spread at a 5 cm × 5 cm Nytran membrane upon MS solid medium plate (8 g/L agar), and the plates were cultured in dark at 22˚C for 4 d. The membrane was then moved upside down on Plich medium (0.5 g/L $KH_2PO_4$, 0.25 g/L $MgSO_4.7H_2O$, 1 g/L Asparagine, 1 mg/L Thiamine, 0.5 g/L Yeast extract, 10 mg/L β–sitosterol, 25 g/L Glucose, 15 g/L agar) with 1.5 mg/L G418 and 300 mg/L timentin, and plates were cultured at 18˚C for 4 d, the membranes were then removed and germinated mycelia were supposed to be observed, the plates were keep in 18˚C for 4 d. Melt rye-sucrose medium with 3 mg/L of G418 was then covered on Plich medium plates for further selection, the G418 concentration could be increased up to 5 mg/L at this stage.

### *Agrobacterium* meditated transformation of *P. viticola* zoospores

AMT of *P. viticola* zoospores followed previous published method on cucurbit powdery mildew pathogen *Podosphaera xanthii* and summarized in S5A Fig [48]. Briefly, culture of *A. tumefaciens* strain EHA105 with pV1F and proper T-DNA construct was prepared following the same methods that described in AMT of *P. infestans*. Separately, the sporangia from *P. viticola* BS5 inoculated grape leaves were harvested by immersion of infected tissues in 30 mL of sterilized water with 0.01% Tween-20 and keep in room temperature for 1 h until zoospores released. About 10 mL (about $1 \times 10^6$ zoospores) *P. viticola* zoospores were gently mixed with same volume of *A. tumefaciens* suspension and co-cultivated for 1 h at room temperature in the dark in an orbital shaker at 65 rpm. Zoospores were then centrifuged for collection (265 g for 5 min), and zoospores in 1 mL of residual were deposited in young detached grape leaves (must be Zitian Seedless, A17 for the best inoculation rate). Two days after inoculation, the grape leaves were rinsed with 5 mg/L G418 and 300 mg/L Timentin to kill *A. tumefaciens* and select *P. viticola* transformants. To be noted, rinsing infection samples with G418 (5 mg/L) and Timentin (300 mg/L) should have no visible negative effect on grape leaves during the above-mentioned procedure. To acquire the stable transformant of *P. viticola*, the obtained isolate was sub-inoculated at zoospore stage on young grape leaves without G418 treatment. After at least three rounds of sub-inoculation, the *P. viticola* isolate that showed stable GFP signal, was defined as a stable transformant.

### Virtualization of GFP

CLSM (Leica AF6000 modular microsystems) was used to take pictures of *P. infestans*. A 489-nm line from an argon ion laser were used to excite green fluorescent protein (GFP). For each assay, six independent leaves were observed for each experiment and each experiment has at least 3 repeats.

### sgRNA design and cloning

CRISPR/Cas12a targets for *MADS-box* (PITG_07059) and *Avr8* (PITG_07558) were designed with overall consideration based on output data from EuPaGDT, CRISPOR and Deep-Cpf1 [79–81]. The used crRNAs were carefully inspected with sequencing data of *P. infestans* JH19

and 1501 to avoid off-target events. DNA oligonucleotides contains crRNAs and direct repeats (DR) were artificial synthesized and cloned into pLY40 based strategy described above and in S6C and S7C Figs.

### Detection of target gene editing

Genomic DNA (gDNA) of *P. infestans* strain or relative transformants was first isolated from liquid cultural mycelia. Specifically, 5 mycelia discs of each *P. infestans* strain from rye-sucrose medium plates were cut and placed into 10 mL of PEA broth, and mycelia for gDNA extraction were harvested after 5 days growth in dark at 18˚C. Mycelia were then blot up by filter paper and gDNA was extracted with Omega E.Z.N.A. Plant DNA Kit (HP). To confirm editing, primer pair F: 5'-ATGGGCCGCAAGAAGATCCAG-3' and R: 5'-TCAAACAGCCACA CGTTGACGCTTG-3' were for checking *MADS-box* editing in *P. infestans* JH19 derived transformants, and primer pair F: 5'-ATGCGCTCAATCCAACTTCTG-3' and R: 5'-TTAC GATGTTTTCGCTTCTTTAAAAAG-3' was for checking *Avr8* editing in *P. infestans* 1501 derived transformants.

### Protein analysis

*P. infestans* protein assay referred to previously described method [12]. Briefly, mycelia were collected from PEA broth culture and total protein was extracted with Beyotime RIPA P0013B Lysis Buffer. Immunoblots were performed as described [78]. Total protein was eluted in sodium dodecylsulfate (SDS) sampling buffer and proceed to western blot analysis. LbCas12a was detected by immunoblotting with anti-LbCas12a (Cpf1) antibody (Sigma/SAB4200777, dilution 1:4000), followed by a secondary antibody conjugated to FITC (ThermoFisher Scientific, dilution 1:5000).

### Southern blot analysis

For T-DNA integration analysis, we performed southern blot with genomic DNA obtained from *P. infestans* transformants or wild-type (WT) isolates. Four μg genomic DNA was digested with restriction enzyme *HindIII* (*Takara*), and separated on DNA agarose gel (1%) and then blotted on positively charged nylon membranes. A *nptII* gene fragment was used as probe. Further preparations of probes DIG-labeling, hybridization and chemiluminescent detection were conducted according to the operation protocol of DIG-High Prime DNA labeling and Detection Starter Kit (ROCHE/11585614910).

### Leaf inoculation with *P. infestans*

For zoospores inoculation assays, zoospores of *P. infestans* strains were collected from plates followed by the same method described above. 10 μL drops with 200,000 to 400,000 zoospores per mL were inoculated on detached leaves from 4 to 6 weeks old potato plants of wild type Désirée and *R8* trangenic potato lines. For mycelial disc inoculation assays, detached potato leaves were inoculated with mycelial discs (5 mm diameter) taken from the edge of 7 days-old rye-sucrose medium culture of *P. infestans*. Inoculated leaves were kept in a plastic tray (30 × 44 × 8 cm) covered with polypropylene film at 22 ± 3˚C. Results were recorded 5 days post inoculation. For each inoculation assay, 8 independent leaves were used for each experiment and each experiment has at least 3 repeats.

## Supporting information

**S1 Table. Primers used in this study.**
(XLSX)

**S2 Table. Plasmids used in this study.**
(XLSX)

**S3 Table. Sequence similarity data used in Fig 1A.**
(XLSX)

**S4 Table. Comparisons of modified AMT protocol and previous reported transformation protocols for _P. infestans_.**
(XLSX)

**S1 Fig. Translocating AtVIP1 from _A. tumefaciens_ to host cells.** (a) The expression cassette used for translocating AtVIP1 fused with GFP (b) Confocal microscopy observation of _N. benthamiana_ leaves infiltrated with _A. tumefaciens_ EHA105 carrying the construct described in (a). Images were taken at 3 days post infiltration. Images are single confocal sections and are representative of images obtained in three independent experiments. White arrows indicate observed nuclei in tobacco cells. Scale bars = 40 μm. Three independent experiments were performed for each assay with similar results.
(TIF)

**S2 Fig. AtVIP1 enhances transient AMT efficiency in wheat tissues.** (a) Schematic representation of plasmid constructs used in this experiment. The pWMB110-gus construct contains a β-glucuronidase expression cassette, carrying the maize _adh1_ intron, in the T-DNA region. (b-c) Transient transformation on wheat leaf and root segments. Dissected wheat tissue segments were inoculated with _A. tumefaciens_ EHA105 carrying the binary plasmid pWMB110-_gus_ and either pV1F or the control plasmid pEV. At 3 days post inoculation, GUS activity was analyzed by histochemical staining. Scale bars = 2 mm. At least 50 leaf or root segments were recorded in each experiment. Each experiment was repeated 3 times and representative results were presented. (d) Quantification of root segments that expressed the _gus_ gene in (c). Statistical differences among the samples were analyzed with Šídák's multiple comparisons test (P< 0.0001: ****).
(TIF)

**S3 Fig. Schematic outline of modified AMT for _P. infestans_.**
(TIF)

**S4 Fig. Western blot characterization of _P. infestans_ HB1501 transformants obtained using pLY40-_gfp_ and pV1F.** Total protein samples were purified from 14 transformants with a GFP signal (up) and 14 transformants without a GFP signal (down). All blots were probed with an anti-GFP antibody. The protein blot was stained with Ponceau S to confirm equal loading.
(TIF)

**S5 Fig. Obtaining _Plasmopara viticola_ BS5 transformant T1, expressing _gfp_, using the optimized AMT method.** (a) Schematic outline of the optimized AMT method for _P. viticola_ BS5. (b) AMT with only pLY40-_gfp_ produced no G418 resistant transformants of _P. viticola_ BS5 (left), while AMT with pLY40-_gfp_ and pV1F produced the transformant T1 (white arrow) that is resistant to G418 (right). Scale bars = 2 mm.
(TIF)

**S6 Fig. The gRNA target regions used for *MADS-box* genome editing in *P. infestans*.** (a) Genome-wide allele ratio analysis of *P. infestans* JH19. (b) Copy number of PITG_07059 (*MADS-box*) relative to single-copy control gene (= 1.0), determined based on read depth in DNA library of JH19 strain. (c) Schematic representation of the constructs used in this experiment. The pLY40-*MADS-box-KO* with either pV1F or pEV were used for *P. infestans* transformation in this experiment. Two gRNAs for *MADS-box* editing are named as g1 and g2. (d) Gene sequence of PITG_07059 (*MADS-box*). Sequences marked in red are targeted by g1 and g2. (e) Sequencing chromatograms of *MADS-box* in wild type JH19, T8 and T16. Both T8 and T16 showed single peaks in either the g1 (left) or g2 (right) target sites. The wild type sequences with gRNA targets are shown at the top of the panel; black arrows indicate the 5' border of the detected deletion.
(TIF)

**S7 Fig. The gRNA regions used for *Avr8* editing in *P. infestans*.** (a) Genome-wide allele ratio analysis of *P. infestans* HB1501. (b) Copy number of PITG_07558 (*Avr8*) relative to single-copy control gene (= 1.0), determined based on read depth in DNA library of HB1501 strain. (c) Schematic representation of the constructs used in this experiment. (d) Gene sequence of PITG_07558 (*Avr8*). Two selected gRNA target regions are marked in red. (e) Sequencing chromatograms of *Avr8* in T3, T10 and T22 of HB1501. Both T3 and T22 showed single peaks in both g1 and g2 target sites. The wild type sequences with gRNA targets are shown at the top of the panel; black arrows indicate the 5' border of the detected deletion.
(TIF)

**S8 Fig. Vegetative growth of *P. infestans* strains.** (a) Mycelia cultured on the rye-sucrose medium were photographed at 5 days post inoculation. (b) Quantification of mycelium diameter of *P. infestans* strains in (a). All data represent average values from three independent experiments with the indicated standard deviations. (c) Microscopy images of the opposite side of the inoculated region of detached potato leaves in Fig 5E show the details of mycelia generated during infection. Images were taken at 5 days post inoculation. White arrowheads indicate the observed sporangia. Scale bars = 1 mm.
(TIF)

**S1 Appendix. Sequences of pLY40, pLY40-gfp, pLY40-Cas12a-gfp.**
(DOCX)

## Acknowledgments

We thank Dr. Howard Judelson (UCR, USA) and Prof. Yuanchao Wang (NAU) for the discussion on *Phytophthora* genome editing. Potato *R8* transgenic line was kindly provided by Dr. Jack Vossen (WUR, Netherlands). The *Plasmopara viticola* BS5 was a kind gift from Prof. Linfei Shangguan (NAU). We thank Dr. Vitaly Citovsky from Stonybrook University for supportive discussion on the original project design. We thank Ms. Ying Zheng (NAU) for confocal microscopy. We thank Ms. Zhao Hu, Dr. Xinyu Liu and Dr. Changling Mou (NAU) for their help with Southern blot analysis and constructive comments on this study.

## Author Contributions

**Conceptualization:** Luyao Wang, Sanwen Huang, Suomeng Dong.

**Data curation:** Luyao Wang, Fei Zhao, Haohao Liu, Suomeng Dong.

**Formal analysis:** Luyao Wang, Fei Zhao, Haohao Liu, Suomeng Dong.

**Funding acquisition:** Luyao Wang, Sanwen Huang, Suomeng Dong.

**Investigation:** Luyao Wang, Fei Zhao, Haohao Liu, Suomeng Dong.

**Methodology:** Luyao Wang, Fei Zhao, Haohao Liu, Han Chen, Fan Zhang, Vladimir Nekrasov, Suomeng Dong.

**Project administration:** Luyao Wang, Sanwen Huang, Suomeng Dong.

**Resources:** Luyao Wang, Sanwen Huang, Suomeng Dong.

**Software:** Luyao Wang, Fan Zhang, Sanwen Huang, Suomeng Dong.

**Supervision:** Luyao Wang, Sanwen Huang, Suomeng Dong.

**Validation:** Luyao Wang, Sanwen Huang, Suomeng Dong.

**Visualization:** Luyao Wang, Suomeng Dong.

**Writing – original draft:** Luyao Wang, Sanwen Huang, Suomeng Dong.

**Writing – review & editing:** Luyao Wang, Han Chen, Fan Zhang, Suhua Li, Tongjun Sun, Vladimir Nekrasov, Sanwen Huang, Suomeng Dong.

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
