## [Decision Letter · Decision Letter 0]

22 Jul 2022

Dear Suomeng,

Thank you very much for submitting your manuscript "A robust Agrobacterium-mediated transformation and genome editing system for Irish potato famine pathogen" for consideration at PLOS Pathogens. As with all papers reviewed by the journal, your manuscript was reviewed by members of the editorial board and by several independent reviewers, all of whom are experts in oomycete transformation. The reviewers indicate that many aspects of the manuscript are not novel, in that CRISPR-cas12 has been used to edit genes in *Phytophthora infestans, *AMT has been used for *Phytophthora *transformation, and functional characterisation of Avr8 and the MADS box transcription factor have been reported. The reviewers were not convinced that enough data were provided to clearly demonstrate an improvement in transformation efficiency compared to previous published methods for Phytophthora, including AMT. The reviewers feel (and I agree) that, as a method, there is a need to better demonstrate the potential broader impact of the VIP1 helper plasmid for AMT of recalcitrant organisms, perhaps including other oomycetes or fungi as indicated by reviewer *2. *In light of the reviews (below this email), we would like to invite the resubmission of a significantly-revised version that takes into account the reviewers' comments.

We cannot make any decision about publication until we have seen the revised manuscript and your response to the reviewers' comments. Your revised manuscript is also likely to be sent to reviewers for further evaluation.

Sincerely,

Paul Birch

Associate Editor

PLOS Pathogens

Bart Thomma

Section Editor

PLOS Pathogens

Kasturi Haldar

Editor-in-Chief

PLOS Pathogens

orcid.org/0000-0001-5065-158X

Michael Malim

Editor-in-Chief

PLOS Pathogens

orcid.org/0000-0002-7699-2064

Reviewer's Responses to Questions

**Part I - Summary**

Reviewer #1: In this MS, the authors claim to present a new and improved methodology of P. infestans transformation with Agrobacterium tumefaciens. In short, the authors show that expressing a host factor (AtVIP1) in Agrobacterium, fused with a Type IV translocation signal, enhances transformation efficiency in P. infestans and a series of recalcitrant (crop) species. Importantly, they show that this improved AMT method can be employed to target and modify P. infestans genes, opening up new avenues for tool development in

Overall, the authors present the data clearly and logically. The figures are well-constructed, and the data is of high quality (some comments on the figures are listed below). There are, however, issues with this MS that I have listed below.

Major Comments:

Whilst the work is important and novel, the advances made are limited. AMT is an established protocol in Phytophthora, and this system (including gene knockout) works in other Phytophthora pathogens (P. palmivora, Tian et al., 2020). The central crux of this work is that the authors claim to have a more efficient transformation system when AtVIP1 is expressed and deployed by Agrobacterium. The questions thus are: (i) Are the claims justified, and (ii) do the claims warrant publication in this journal?

In short, i think the conclusions are justified though I am not convinced that the advance represents a major leap forward.

Reviewer #2: The main new discovery of this paper is that inclusion of a helper plasmid expressing AtVIP1 increases agrotransformation rates of Phytophthora infestans by about ten-fold. VIP1 is a plant bZIP protein shown previously to be involved in Agrobacterium-mediated transformation (AMT) in plants. Otherwise, the P. infestans transformation system described here is adapted from a method used by Vijn and Govers in 2003 in P. infestans, Phytophthora palmivora, and Pythium ultimum, and which was later adapted by others to P. sojae and in modified form to P. palmivora.

A survey of the literature identified four methods for transforming Phytophthora: protoplast transformation, zoospore electroporation, AMT, and microprojectile bombardment. It is impossible to say which is best, since the number of laboratories involved are limited and several different species are involved. Suffice it to say that the community benefits from having options.

In this paper, the authors first test the concept of using a VIP1 helper plasmid. First, they demonstrate that a VIP1-VirF-gfp fusion containing a type 4 secretion sequence (T4SS) was translocated into Nicotiania. Then, they demonstrated that adding this construct increased transient expression of GUS by about 10-fold in wheat, a species that is known to be difficult to transform. They then apply the method to P. infestans, obtaining data indicating that up to a 10-fold increase in transformation efficiency was achieved. Then using AMT with the helper plasmid and a CRISPR-Cas12a construct similar to that described previously for P. infestans (reference 12), they edit a MADS-box transcription factor (suppressing sporulation) and RXLR gene Avr8 (allowing evasion of resistance gene R8). The phenotypes resulting from editing the MADS gene are as expected since a paper from 2019 described the same phenotype in P. infestans using homology-based silencing. The Avr8 result is consistent with results from genetic segregation and agroinfiltration studies that demonstrated its interaction with R8.

My overall impression of the paper: The modified AMT method is a useful addition to the existing methods for P. infestans transformation. The CRISPR-Cas12a results are nice to see but not remarkable based on the prior paper on Cas12a in P. infestans, and the prior publications on the function of MADS and Avr8, although it is nice to see the Avr8-R8 interaction supported by a third type of evidence (adding to the segregation and agroinfiltration studies). Depending on what numbers you refer to, this either increases transformation rates 10-fold (based on data in the paper) or 4-fold (based on comparison to the existing AMT paper for P. infestans. Whether the modest increase claimed is sufficient for publication in PLOS Pathogens is an editorial decision. It is not clear if the method is more efficient than other transformation methods. However, I recognize that there could be major impact of the VIP1 helper if applied to other oomycetes and non-oomycetes as well.

Reviewer #3: Wang and colleagues report an Agrobacterium-mediated transformation and genome editing system for the Irish potato famine pathogen. This study comes after two publications on the same topic (Vijn & Govers, reference 7; Wu et al., reference 20). While the manuscript explores an interesting idea that has the potential to optimise the genetic manipulation of Phytophthora infestans, it lacks a comprehensive evaluation of transformation efficiency and a careful comparison to other methods using different P. palmivora strains.

**Part II – Major Issues: Key Experiments Required for Acceptance**

Reviewer #1: In my view, the presence of pV1F does have a demonstrable impact on transformation rates. Whilst the primary gene constructs may have an effect (CAS12a transformation experiments seem to have lower yields), there is a consistent and significant positive impact on transformation outcomes.

What is not clear, however, is whether there is a tangible effect on the number of transformants that express the gene of choice (GFP or CAS12a). Similarly, it is unclear whether improved transformation efficiency leads to a higher proportion of transformants carrying multiple inserts (resulting from successful integration into the chromosome). These characteristics (expression, integration, and insert number) are pertinent to transformation experiments and subsequent (genetic) analyses. Yet, the authors do not report on these features. I think they should; as with that data, the MS would greatly interest the reader.

Reviewer #2: The paper would have a much greater impact if the concept of using the VIP1 helper plasmid was extended to other species, including fungi where AMT is common but not always efficient. Such information could be a real breakthrough for many pathogen species.

(One could also argue that the use of the VIP1 helper plasmid to enhance transformation in wheat is of equal, if not greater, impact; it is somewhat odd to see those results mixed in with the Phytophthora data but inasmuch as that data are shown as Supplementary Figures I do not object. I am surprised that this approach was not tested earlier in many plant species. I could only find a few papers on the subject, including reference 26 which describes how VIP overexpression increased transformation rates in Nicotiana. Perhaps dicot bZIPs are conserved such that orthologues can provide a VIP1-like activity, whilst monocots are more diverged.)

For a paper that focuses mostly on methods, the transformation protocol was lacking in detail. These are some of the examples: Methods is fairly silent on how zoospores were generated and only in the Discussion are the actual amounts of zoospores used mentioned. The age of the cultures used to generate zoospores were 7 to 14 days in line 430, but 14 days in Fig S3. Is one better than the other? Rather than giving the details, we are just told that "the proper antibiotics" were used (li 432). Centrifugation speeds/g-forces were not provided. Agar concentrations in the overlay were not described. Also, whilst Methods says that Nytran was used, Fig. S3 alternates between Nytran and Hybond N+. Is there is difference?

Whilst the paper does convincingly show that the helper plasmid raises transformation rates, the transformants themselves are poorly characterised. Perhaps this is not necessary here since there have been several papers on AMT in P. infestans and relatives already. However, as cited on li 345 about 1/3 of AMT events in Arabidopsis result in multiple integration events, so should not issues like this also be tested here? It would also be useful to address the issue of heterokaryons or mixtures arising from the method.

Since the authors are promoting their AMT method as an improvement over other transformation methods, could they report on the number of their GFP transformants that expressed easily visible levels of fluorescence, or the stability of expression? Low expression could be a problem with single-copy transformants.

Considering that the authors are promoting their AMT method as an improvement over past AMT methods, could they discuss the variables that they tested as part of optimising their procedure?

Reviewer #3: - The central idea of this manuscript is to improve transformation rate by ectopic expression of a plant gene (AtVIP1) in A. tumefaciens. Conceptually, this is not new as it has been already used in various systems. For the specific purpose of oomycete transformation, the authors did not compare their method with the previously published AMT protocol for P. infestans. Instead, they report 61 and 22 transformants for two strains but the amount of zoospores is not mentioned in the M&M section, making it impossible to compare with the efficiency reported by Vijn & Govers. I would expect the authors to provide a comparison table using PEG-protoelectroporation, AMT and AMT+AtVIP1

- On a similar note, the authors use sequence alignments (blast) to claim that oomycetes lack homologues of specific plant proteins important for AMT. Overall, this just reflects the phylogenetic distance between oomycetes and plants, and does not mean that the activities are absent from oomycete genomes. Did the authors consider domain-based screening and 3D structure modelling?

- The sentence "The first AMT protocol of P. infestans was established in 2003 by Ah-Fong et al" (line 93) is incorrect. The first protocol was indeed established in 2003 but the correct reference is Vijn & Govers, Molecular Plant Pathology (doi: 10.1046/j.1364-3703.2003.00191.x). The reference number (7) at the end of the sentence is correct.

- The effect of ectopic expression of AtVIP1 in A. tumefaciens on wheat transformation is not related to the manuscript main focus (P. infestans). As pointed out in the introduction, improvement of plant transformation by ectopic expression of plant genes in A. tumefaciens has already been established.

**Part III – Minor Issues: Editorial and Data Presentation Modifications**

Reviewer #1: Minor comments:

Figure 1. The phylogenetic tree looks superficial and confuses me. Does the tree mean to depict evolutionary relationships of specific genes? If so, i am not clear which genes were used here and why the tree is branched the way it is. It is also unclear how the values were calculated (it is described in the methods and results, but I am unfamiliar with the method).

Figure 2. Panel d: Please include a negative control (transformed with EV) to demonstrate that signal is specific to GFP (not fluorescence coming from mycelia). Panel e: Please have a coomassie stained gel as a loading control; indicate the (expected) size of the GFP band.

In figure 3b, please add a non-GFP control and, for 3c, a loading control (as specified for fig 2).

Finally, the text does have a fair number of spelling and grammatical errors. This MS should undergo a solid round of editing to ensure it meets publication standards.

Reviewer #2: In Introduction and Discussion, the authors make many statements about problems with the existing methods for transformation. The authors are pleading why their method is a major improvements, but their arguments are too strong, not always justified, and include errors. For example:

Line 66: Transformation rates cited in papers 20 or 30 years ago ("0.1 to 2 per microgram") may not hold today as improvements have likely been made over time including different enzyme mixtures and osmotica. Moreover, there is no way to compare those rates with the AMT rates described here. The methods are totally different.

Line 67: The statement that most transformants from protoplast transformation are heterokaryons is wrong. I could only find a statement in the literature saying that heterokaryons were NOT found. Furthermore, the current paper does not test for heterokaryosis.

Line 67: References 5 to 7 are used to describe the protoplast transformation method of obtaining stable transformants. However #5 does not involve stable transformation whilst #7 involved AMT

Li 68: The statement that "Microprojectile projectile bombardment and zoospore electroporation seem feasible, but a low transformation efficiency still restrains related research (8,9)." This is odd since the number of transformants in paper #8 exceeds those in the current paper. The number of transformants from the optimised protocol in paper #9 is nearly identical to those described in the current paper.

Li 69: Whilst phenotypes may vary between transformants obtained in past studies, perhaps due to random integration sites, would not this also be true for AMT transformants? Furthermore, I did not see any data in the paper that thoroughly evaluated the phenotypes or integration events of the AMT transformants.

Li 77: Since the number of transformants in the Cas12 experiments described in reference 12 exceed that of the current AMT paper, it is not obvious that the transformation efficiencies from the protoplast method were low.

Li 320: The authors spend a lot of time presenting calculations that show that their method is 4 times more efficient than the AMT method described in ref. 7. Is this necessary? Please, a sentence focussed on the data in Table 1 would be sufficient.

Li 331: The authors indicate that in the protoplast method, several problems might result including integration into an unstable site, degradation of the DNA by cellular enzymes. But would not similar events result from AMT? I am very familiar with AMT in plants, where T-DNA expression can also be unstable, or where only part of the T-DNA might integrate.

Li 336: Here again the authors appear to be claiming that their AMT method is less likely to result in heterokaryons than the protoplast method. Do the authors have data that show that many protoplasts used for transformation have multiple nuclei, resulting in heterokaryons? One could argue that heterokaryosis may be more common in AMT since transformation may occur as zoospores germinate and become multinucleate. Moreover, a fraction of P. infestans zoospores are multinucleate (for example, see Protoplasma 135:173, 1986). Whilst some papers may describe single-spore purification, this is just good microbiological practice to exclude mixed strains, and not just a way to eliminate potential heterokaryons. Based on the filter selection method shown in Fig. 2b I suggest that single-sporing should be part of the AMT protocol for P. infestans.

I suggest that Figure 1a be moved to Supplementary material. It is mostly relevant to studies of plants, not Phytophthora. One could also argue that for the broad readership of PLOS Pathogens, non-oomycetes should also be included. Nevertheless, I am not sure of the value of the figure it is not surprising that weak similarity was observed for VIP1 in oomycetes since transcription factors typically show little similarity between kingdoms outside of the DNA-binding domain, which represents only a small part of the protein.

I also did not like Fig. 1b. It does not add much to my understanding of the procedure, and seems out of context as it is introduced before the actual AMT experiments with P. infestans.Li 347: Whilst the use of A1A2 (self-fertile) strains of P. infestans to obtain biallelic edited strains through selfing is intriguing, have the authors tried this? Are A1A2 strains polyploid, making them difficult targets for gene editing? And are there data on how homogeneous phenotypes would be in the resulting strains?

Li 359: The paper that silenced the P. infestans MADS box gene is cited here in Discussion, but why was this not cited earlier instead of referencing the P. sojae MADS gene?

Li 375: This states that "no direct evidence has ever been presented" about whether Avr8 is involved in resistance mediated by R8. Direct evidence, however, is in the paper by Vossen et al (Theoretical and Applied Genetics 129: 1785, 2016). It is curious that this was not cited in the manuscript. A similar experiment confirming the interaction between Avr8 and R8 (by transient expression in Nicotiana was published in J. Exp Bot. 69, 1545, 2018. Avr8 (PITG_07558), under the name AvrSmira2, was also implicated as interacting with R8 based on segregation studies in reference 44. On this topic, on li 373 did the authors mean "its responses to the R gene" instead of "its responses to the effectors."

Fig. 4g: might the reduced lesion size in the MADS-edited strains be the result of fewer infection events, since sporangia were absent? Here infections were somehow done with mycelia, which is unnatural. Methods only provides details of how zoospore inoculations were performed.

Li 156: some explanation for the inclusion of VirF sequences would help the reader.

Reviewer #3: line 48, 254: misspelled "avirulance"

line 57: missing space

line 103: "a set of proteins from host" maybe "a set of host proteins"?

line 191: "in both two testedconstruct" missing space, and use either "two" or "both"

PLOS authors have the option to publish the peer review history of their article (what does this mean?). If published, this will include your full peer review and any attached files.

Reviewer #1: **Yes: **Edgar Huitema

Reviewer #2: No

Reviewer #3: No
---

## [Decision Letter · Decision Letter 1]

3 Feb 2023

Dear Dr. Dong, Dear Suomeng

Thank you very much for submitting your manuscript "A modified Agrobacterium-mediated transformation and genome editing system for Irish potato famine pathogen" for consideration at PLOS Pathogens. As with all papers reviewed by the journal, your manuscript was reviewed by members of the editorial board and by several independent reviewers. The reviewers appreciated the attention to an important topic. Based on the reviews, we are likely to accept this manuscript for publication, providing that you modify the manuscript according to the review recommendations.

In particular, reviewer 3 (but all reviewers to some extent) recommends changing the title and emphasis in the paper to incorporate a major advance - AMT working for an obligate biotrophic filamentous pathogen. I concur with, as it will help to attract more attention to a breakthrough in your work, given the prevalence of obligate biotrophs challenging crop production. Please pay attention to all of the comments, major and minor, from reviewer 2. All of these changes are editorial and will improve the manuscript in my opinion. 

Sincerely,

Paul Birch

Academic Editor

PLOS Pathogens

Bart Thomma

Section Editor

PLOS Pathogens

Kasturi Haldar

Editor-in-Chief

PLOS Pathogens

orcid.org/0000-0001-5065-158X

Michael Malim

Editor-in-Chief

PLOS Pathogens

orcid.org/0000-0002-7699-2064

Reviewer Comments (if any, and for reference):

Reviewer's Responses to Questions

**Part I - Summary**

Reviewer #1: This MS is a revised version and describes the impact of VIP1, a protein that, when expressed in eukaryote cells, enhances Agrobacterium tumefaciens-mediated transformation (AMT). The authors claim that the expression of VIP1 in A. tumefaciens leads to enhanced transformation rates in wheat, P. infestans and the obligate biotroph Plasmopara viticola.

The authors have made significant changes to the MS, including P. viticola transformation. The data is not overly convincing (one P. viticola transformant was presumably generated in one experiment), but considering the technical challenges associated with manipulating obligate biotrophs, this is a significant achievement. I am convinced that this approach will be tried and tested in other pathosystems, including obligate biotrophs.

Unfortunately, the authors opted not to include data describing cassette integration and copy number. While it would have been a nice addition, it does not diminish the overall findings and message of the work.

Taken together, this work represents a significant advance, possibly eliminating barriers that prevent transformation and gene editing in Eukaryote organisms considered recalcitrant to genetic transformation and modification.

Reviewer #2: This is a revised version of a paper that mostly describes an improved Agrobacterium-mediated transformation procedure (AMT) for the plant pathogen Phytophthora infestans; the paper also goes on to demonstrate that the method can also be used to express a Cas12a protein for gene editing. That expression of the Arabidopsis VIP1 protein increased transformation frequencies in Phytophthora is convincing. I can see from the responses to reviewers that many improvements have been made compared to the original manuscript. It was disappointing that a test of the VIP1 strategy in fungi was not attempted, as this would have greatly increased the scope and impact of the paper, changing it from a quantitative improvement over the prior AMT method to something with broad application. Curiously, whilst the paper also reports that a similar strategy aided for wheat transformation--a finding that would have major impact--this is not mentioned in the Abstract. Nevertheless, it was exciting to see that the method apparently enabled the transformation of an obligate pathogen, the downy mildew Plasmopara viticola; this finding broadens the impact of the paper.

Whilst the paper is improved in many ways, there are still some problems and room for improvement. Issues related to the science are listed in points 1 to 17. The paper also needs improvement in some of the wording; some examples are shown after point 17.

Reviewer #3: This is a resubmission of a manuscript by Wang and colleagues about a modified Agrobacterium-mediated transformation for the Irish potato famine pathogen. The authors have made significant effort to address my concern. However, with the additions made in the revised version, I am now facing a dilemma: on one side, I am much more supportive to the manuscript. On the other side, I believe the Plasmopara viticola transformation is the most impactful finding reported in this study. It should be better emphasized and should be part of the title. I therefore expect the authors to reconsider the writing of their manuscript toward P. viticola transformation. I am also curious about possible toxicity effect of G418 on plant leaves. I would like to stress the point that such a protocol would be of invaluable interest for other researchers working on obligate biotrophs, including Bremia lactucae.

**Part II – Major Issues: Key Experiments Required for Acceptance**

Reviewer #1: none

Reviewer #2: 1. Whilst I am glad to see a Southern blot used to characterize the nature of the transformation events, it is impossible to interpret the results since none of the plasmid maps show the full plasmids, location of HindIII sites, etc. Still, why do five of the seven transformants show the same lower band? (I can not interpret this result without maps and smaller size markers on the gel). It would also have been better to use the whole plasmid as a probe rather than npt2.

2. The results on Plasmopara viticola are important but preliminary, since the nature of the transformants (copy number, etc) are preliminary. As I think that transformation of this species is one of the more important parts of the paper, it is unfortunate that the results are so slim. Also, is it clear that the VIP1 plasmid was needed for transformation? This is not a trivial question considering that AMT without VIP1 has been used previously against many fungi include a powdery mildew, which like P. viticola is an obligate pathogen. Also, when the authors refer to the P. viticola transformants being stable for three generations, was this tested in the absence of selection. Methods does not make it clear how the stability test was performed.

3. Li 304. I am confused by the plant infection experiments with the MADS-edited strains. Methods states that the infections were performed using zoospores, but how could this be possible for strains that do not make sporangia? As written, it seems that infection did not occur but the mycelia grew poorly.

4. Fig. 1b: This diagram may not represent how transformation is occurring. It seems unlikely that transformation is occurring on flagellated zoospores. The treatment conditions are likely to cause encystment of the zoospores followed by emergence of a germ tube. So the figure might be deceptive. This also raises the point of when transformation (DNA integration) is actually occurring. If this occurs after nuclear division begins post-germination, the transformants could be heterokaryons. Did the authors test this by passage through a single zoospore stage?

Reviewer #3: No more experiment needed, but manuscript rewriting to be considered.

**Part III – Minor Issues: Editorial and Data Presentation Modifications**

Reviewer #1: none

Reviewer #2: 5. Li 373 to 390: This part of Discussion needs to be improved. It talks about limitations of the protoplast method followed by limitations of AMT. But some of the points used for one could also apply to the other, such as the possibility of chimeric mycelia (there are no data to indicate that this is the case with AMT).

6. Li 213: do the authors mean the Bremia lactucae RPL41 promoter, or the P. sojae promoter (from whence the plasmids originated). On a similar topic, many of the components in the plasmids in Fig. 2 and elsewhere are not defined in the figure legends or main text.

7. Li 208: Not clear what the authors mean by "poorly amenable, considering the many papers that describe the use of transformation. Also, references 6 and 31 do not address this issue.

The content of the first and much of the second paragraph of Discussion are odd. These paragraphs are focused on plant genetics, not the main topic of this paper (oomycetes).

8. Li 86. Both stable and transient silencing have been used.

9. Li 108. It would be appropriate to describe the broad hosts against which AMT has been used; in the context of Plos Pathogens some mention of the extensive use of AMT against fungi would be appropriate.

10. Li 116. It is hard to understand the point being made about low numbers of zoospores in some strains. What the concentration of zoospores were in the prior AMT paper for P. infestans is unclear. Perhaps more importantly, It would be surprising that anyone would want to work on such strains since it would be hard to perform plant infections. Maybe the authors should instead focus on oomycete species that do not make zoospores?

11. Li 159. "Homologous" is misused. The approach is measuring similarity, not homology.

12. Table 1: The headers would be less confusing if they said GFP+/G418r (adding the plus).

13. Line 384. The discussion of methylation seems odd considering that the senior author (plus others) have previously reported a dearth of cytosine methylation in Phytophthora.

14. Li 35: I question the use of the word "unique". Compared to what? One could say that anything is unique.

15. Li 36-38, 61-63. I question the use of "poorly understood". Whilst many questions remain, we are past the use of "poorly". Consider other wording such as "much remains to be learned."

16. Li 70. I question the use of the word "non-culturable". It can be cultured on leaves. It would be more precise to say that it can not be cultured on axenic media.

17. Li 79. Better to say "well-understood" rather than "well-studied"; the latter implies that people did not do a good job.

OTHER WORDING ISSUES

There are many poor word choices, grammar, or spelling in the paper, so the manuscript should be checked and improved. Here are just a few examples, as there were too many for me to list:

Li 82. Did the authors really mean to use hysteretic?

Li 90. Use of "open-minded" is wrong, but it is fine to say that we need to be open to new methods.

Li 165. Presumably the authors are using "similarity coefficient" in place of "sequence similarity", the latter of which is more commonly applied to BLAST.

Li 453. Might be better to replace "might be required" to "could be used."

Author summary: many of the sentences have problems. For example, what "endogenously" means on line 66 is not clear. Also, Li 72: Has the POTENTIAL to accelerate. It has not yet done this.

Reviewer #3: The authors adequately addressed all my previous comments.

PLOS authors have the option to publish the peer review history of their article (what does this mean?). If published, this will include your full peer review and any attached files.

Reviewer #1: No

Reviewer #2: No

Reviewer #3: No

Figure Files:

Data Requirements:

Reproducibility:

References:

---

## [Editor Report · Decision Letter 2]

6 Apr 2023

Dear Dr. Dong, Dear Suomeng,

We are pleased to inform you that your manuscript 'A modified Agrobacterium-mediated transformation for two oomycete pathogens' has been provisionally accepted for publication in PLOS Pathogens.

Before your manuscript can be formally accepted you will need to complete some formatting changes, which you will receive in a follow up email. A member of our team will be in touch with a set of requests. Please also check carefully to correct spelling throughout - both Phytophthora and Plasmopara are mis-spelled in abstract and author summary, for example. Please also carefully check through and correct spelling mistakes in the reference section.

Best regards,

Paul Birch

Academic Editor

PLOS Pathogens

Bart Thomma

Section Editor

PLOS Pathogens

Kasturi Haldar

Editor-in-Chief

PLOS Pathogens

orcid.org/0000-0001-5065-158X

Michael Malim

Editor-in-Chief

PLOS Pathogens

orcid.org/0000-0002-7699-2064
---

## [Editor Report · Acceptance letter]

19 Apr 2023

Dear Dr. Dong,

We are delighted to inform you that your manuscript, "A modified *Agrobacterium*-mediated transformation for two oomycete pathogens ," has been formally accepted for publication in PLOS Pathogens.

Best regards,

Kasturi Haldar

Editor-in-Chief

PLOS Pathogens

orcid.org/0000-0001-5065-158X

Michael Malim

Editor-in-Chief

PLOS Pathogens

orcid.org/0000-0002-7699-2064